# EpiFormer: Learning Antigen-Antibody Interactions for Epitope Prediction with Geometric Deep Learning

## Abstract

Antibodies neutralize foreign antigens by binding to specific surface regions called epitopes. Computational epitope prediction is critical for understanding immune recognition and guiding antibody engineering. Recent advances in protein structure prediction and geometric deep learning have improved general protein binding site prediction. However, antibody-specific epitope prediction remains challenging due to the need for better protein representations and jointly modeling antibody-specific interaction patterns under severe class imbalance and scarce training data. We propose *EpiFormer*, an encoder-decoder architecture featuring two key architectural contributions: (1) an E(3)-equivariant multi-relational graph neural network (EGNN-R) that unifies geometric equivariance with multi-relational message passing for better protein representation learning, and (2) a parallel cross-attention mechanism coupled with EGNN-R layers to explicitly capture antigen-antibody interactions. We further introduce tailored loss functions with protein language model (PLM) embeddings to mitigate extreme class imbalance and data scarcity. Our method significantly outperforms existing baselines on the AsEP dataset by achieving a 28% improvement in F1 score and 13% improvement in MCC over the previous best method. This work advances the state-of-the-art in epitope prediction and demonstrates the value of antibody-aware geometric modeling for epitope prediction.

## 1. Introduction

Antibodies are Y-shaped proteins that recognize and neutralize foreign substances by binding to specific surface

regions on antigens called epitopes. Accurate identification of epitopes is critical for therapeutic antibody design, vaccine development, and understanding immune recognition (Norman et al., 2020; Joubbi et al., 2024). While traditional experimental approaches for epitope mapping are time-consuming and expensive, computational methods offer the potential for rapid screening of candidate therapeutics (Hummer et al., 2022).

Computational epitope prediction methods can be categorized as *antibody-agnostic* or *antibody-aware*. Antibody-agnostic methods, including DiscoTope3 (Høie et al., 2024), EpiGraph (Choi & Kim, 2024), and GraphBepi (Zeng et al., 2023), predict binding sites as intrinsic properties of the antigen without considering the specific antibody. However, epitopes are fundamentally antibody-specific, as different antibodies targeting the same antigen bind to different surface regions (Liu et al., 2024). Antibody-aware methods such as PECAN (Pittala & Bailey-Kellogg, 2020), MIPE (Wang et al., 2024b), and WALLE (Liu et al., 2024) address this by conditioning predictions on the antibody structure.

However, current antibody-aware methods share a common limitation: they encode antigen and antibody independently and fuse information only at a late stage (*late fusion*). This design cannot capture co-dependent structural features that arise from binding. Antibody CDR loops adapt their conformation to the epitope, and epitope residues similarly adjust to accommodate the paratope. Encoding each chain without knowledge of its counterpart misses these important interaction-specific geometric signatures.

We propose *EpiFormer*, an encoder-decoder architecture that addresses these limitations through two innovations. First, we introduce *EGNN-R*, a multi-relational E(3)-equivariant graph neural network where each edge relation type has its own learned transformation, enabling the model to distinguish covalent backbone constraints from non-covalent spatial contacts. Second, we employ *interleaved and bi-directional cross-attention* at every encoder layer, rather than only at the output, allowing antigen and antibody representations to inform each other throughout the encoding process. We further develop a joint training objective combining Dice loss, bipartite edge prediction, and distance supervision to address severe class imbalance.

[1]Anonymous Institution, Anonymous City, Anonymous Region, Anonymous Country. Correspondence to: Anonymous Author <anon.email@domain.com>.

Preliminary work. Under review by the International Conference on Machine Learning (ICML). Do not distribute.

Our main contributions are:

1. **EGNN-R**, a multi-relational extension of E(3)-equivariant message passing for better protein representation learning, where each edge relation type has its own learned transformation function.

2. **Interleaved cross-attention** that enables antigen-antibody information exchange at every encoder layer, rather than only after independent encoding.

3. State-of-the-art performance on AsEP benchmark dataset (0.433 F1, 0.404 MCC), improving over the best existing method by 28% F1 and 13% MCC.

## 2. Related Work

**Antibody-agnostic methods** predict epitopes without considering which antibody is binding. Early epitope prediction methods relied on sequence features such as hydrophilicity and accessibility (Jespersen et al., 2017). Structure-based approaches, such as DiscoTope3 (Høie et al., 2024), EpiGraph (Choi & Kim, 2024), and GraphBepi (Zeng et al., 2023), improved upon these by incorporating 3D information.

**Antibody-aware methods** condition predictions on the binding counterpart. PECAN (Pittala & Bailey-Kellogg, 2020) encodes antibody and antigen with separate graph convolutions and combines them via bilinear attention. MIPE (Wang et al., 2024b) uses multi-modal contrastive learning to align sequence and structure representations, applying multi-head attention for late-stage fusion. WALLE (Liu et al., 2024) formulates epitope prediction as bipartite link prediction between antibody and antigen residue graphs. EpiScan (Wang et al., 2024a) incorporates CDR masking into sequence embeddings. All of these methods share a *late fusion* architecture in which antigen and antibody are encoded independently before cross-chain attention is applied.

**E(3)-equivariant neural networks** preserve geometric properties under rotations and translations, making them well-suited for molecular modeling. EGNN (Satorras et al., 2021) achieves equivariance by updating coordinates along displacement vectors scaled by learned invariant functions. Equiformer (Liao & Smidt, 2022) and EquiformerV2 (Liao et al., 2023) extend this with SE(3)/SO(2) equivariant attention using spherical harmonics. GearNet (Zhang et al., 2022) introduced multi-relational protein graphs with seven edge types encoding sequential and spatial relationships, but uses a shared message function across relations. Surface-based methods, including MaSIF (Gainza et al., 2020) and AtomSurf (Mallet et al., 2023), operate on molecular surface representations with geodesic or spectral convolutions.

Several methods from adjacent domains share architectural similarities with *EpiFormer*. **CheapNet** (Lim et al., 2025) predicts protein-ligand binding affinity using geometry-informed graph networks (GIGN) for each entity, followed by cross-attention. This is similar to our use of separate encoders with cross-attention, but CheapNet applies cross-attention only at the final layer (late fusion), whereas *EpiFormer* interleaves it at every encoder block. **EquiPocket** (Zhang et al., 2023) uses E(3)-equivariant message passing for ligand binding site prediction but is antibody-agnostic and operates at atom-level rather than the residue-level. **DiffDock** (Corso et al., 2022) employs SE(3)-equivariant score networks for molecular docking, demonstrating the value of equivariant architectures for protein-ligand interactions. **Boltz-1/2** (Wohlwend et al., 2025; Passaro et al., 2025) use Pairformer architectures with triangle attention for structure prediction; Boltz-2 adds cross-attention between binding counterparts, but applies it to pairwise representations rather than node embeddings.

**EpiFormer** differs from these methods by addressing their limitations for epitope prediction. Like CheapNet, we use separate encoders with cross-attention, but apply attention at every layer rather than only at the output. Like GearNet, we use multi-relational graphs, but learn separate transformations per relation type. Similar to EGNN and EquiPocket, we maintain E(3)-equivariance, but extend it to the multi-relational setting. Our key architectural distinction is *interleaved* cross-attention: while prior antibody-aware methods (PECAN, MIPE, CheapNet-style) encode chains independently before late fusion, *EpiFormer* enables cross-chain information flow throughout the encoding process.

## 3. Methods

In this section, we present preliminaries, the architecture of *EpiFormer*, and the customized joint training objective.

### 3.1. Preliminaries

**Graph construction.** The protein 3D structure is described as a point cloud of atoms $\{v_{i,k}\}_{1 \le i \le p, 1 \le k \le p_i}$, where $p_i$ is the number of atoms in residue $v_i$ and $p$ represents the number of amino acid residues in the protein. The first four atoms in any residue correspond to its backbone atoms (N, $C_\alpha$, $C_\beta$, O) and the rest are its side chain atoms. The 3D coordinate of an atom $v_{i,k}$ is denoted as $x(v_{i,k}) \in \mathbb{R}^3$. Since we work with the *unbound* structures or point clouds of antigen(ag) and antibody(ab), we build two completely independent residue graphs $\mathcal{G}_{ag} = (\mathcal{V}_{ag}, \mathcal{E}_{ag}, \mathcal{R})$, and $\mathcal{G}_{ab} = (\mathcal{V}_{ab}, \mathcal{E}_{ab}, \mathcal{R})$. Vertex $v_i \in \mathcal{V}$ represents residue $i$, centered on $C_\alpha$ at coordinate $\mathbf{x}_i \in \mathbb{R}^3$. $|\mathcal{V}_{ag}| = n$, $|\mathcal{V}_{ab}| = m$, and edges $e_{i,j} \in \mathcal{E}$ encode structural/functional relationships between residues.

Each node $v_i \in \mathcal{V}$ is attributed a node feature vector $\mathbf{h}_i \in \mathbb{R}^{d_h}$ and a node coordinate matrix $\mathbf{X}_i \in \mathbb{R}^{3 \times 4}$ consisting of four backbone atoms $\xi = \{N, C_\alpha, C_\beta, O\}$ ($\mathbf{x}_i$ is short for $\mathbf{x}_{i,C_\alpha}$). Specifically, the node feature vector $\mathbf{h}_i$ constitutes handcrafted geometric features and PLM-derived embeddings to capture both structural and evolutionary information. In addition, each edge $e_{i,j}$ is attributed an edge feature vector $\mathbf{f}_{i,j} \in \mathbb{R}^{d_f}$ and a tuple of edge relations $\mathbf{r}_{i,j} \subseteq \mathcal{R}$. The edge vector $\mathbf{f}_{i,j}$ encodes features such as distances and angles to capture both local geometry and global structural context. The set of edge relations $\mathcal{R} = \{\rho_1, \rho_2, \rho_3, \rho_4\}$ captures distinct protein interactions: sequential relations for peptide bonds ($\rho_1$) and short-range coupling ($\rho_2$), plus spatial relations for local packing shells via $K$-nearest neighbors ($\rho_3$) and medium-range contacts within 8 Å ($\rho_4$). Please refer to Appendix A.3 for further details. We extend the notation of these attributes to refer to the residue graph $\mathcal{G}$ of the antigen (or antibody) as $(\mathbf{H}, \mathbf{X}, \mathbf{F}, \mathbf{R})$.

**Problem Formulation.** We formulate the problem as a binary node classification task. In this task, a residue $v \in \mathcal{V}_{ag}$ is labeled as an epitope (1) if it is within 4.5Å of any residue in $\mathcal{V}_{ab}$; otherwise, it is labeled as a non-epitope (0). The classifier predicts the epitope node labels $\hat{y}_{ag}$ using $f: v_{ag} \to \{0, 1\}$ and is defined as:

$$\hat{y}_{ag} = f(v_{ag}; \mathcal{G}_{ag}, \mathcal{G}_{ab}) = \begin{cases} 1 & \text{if } v_{ag} \text{ is an epitope,} \\ 0 & \text{otherwise.} \end{cases}$$
(1)

**Equivariance and Invariance in E(3) Space.** Traditional graph representations of proteins capture connectivity but ignore crucial 3D geometric information. Recently, proteins have been naturally modeled as geometric graphs that encode both topological connectivity and 3D spatial coordinates of atoms. Since molecular properties remain unchanged under rigid body transformations (rotations, translations, reflections), geometric GNNs incorporate E(3)-equivariance as an inductive bias to respect these fundamental symmetries (Jiao et al., 2023).

For a protein with coordinates $\mathbf{X} \in \mathbb{R}^{3 \times m}$ and scalar features $\mathbf{h} \in \mathbb{R}^d$, an E(3)-equivariant function $f$ satisfies:

$$f(g \cdot \mathbf{X}, \mathbf{h}) = g \cdot f(\mathbf{X}, \mathbf{h}), \quad \forall g \in \text{E}(3)$$
(2)

where group actions are defined as translations $g \cdot \mathbf{X} = \mathbf{X} + \mathbf{b}$ or rotations/reflections $g \cdot \mathbf{X} = \mathbf{OX}$ with $\mathbf{O} \in \text{O}(3)$. This contrasts with E(3)-invariant functions, which satisfy $f(g \cdot \mathbf{X}, \mathbf{h}) = f(\mathbf{X}, \mathbf{h})$, producing outputs unchanged by coordinate transformations.

## 3.2. EpiFormer

In this section, we present the architecture of *EpiFormer*, an encoder-decoder framework for antibody-antigen binding-site prediction. The model receives two disjoint multi-relational residue graphs, $\mathcal{G}_{ag}$ and $\mathcal{G}_{ab}$, processes them with independent encoders that produce residue-level embeddings, and then passes these embeddings to a decoder to reconstruct the bipartite adjacency matrix $\hat{\mathcal{E}}_{bg} \in \{0, 1\}^{n \times m}$. A desirable property of our proposed framework is its E(3)-equivariance to address a broader range of symmetries in antigen-antibody interactions and preserve the geometry of these proteins. The overall workflow is presented in Figure 1 while the algorithm is provided in the Appendix 1.

**Encoder.** The *EpiFormer* contains two parallel encoders with no shared parameters, one dedicated to the antigen chain and the other to the antibody chain, as shown in Figure 1 **(a)**. Both encoders operate on heterogeneous residue graphs $\mathcal{G}_{ag}$ and $\mathcal{G}_{ab}$ whose nodes encode Cartesian coordinates $\mathbf{x}_i \in \mathbb{R}^3$, geometric descriptors $\mathbf{h}_i^{geo} \in \mathbb{R}^{d_{geo}}$ and PLM embeddings $\mathbf{h}_i^{plm} \in \mathbb{R}^{d_{plm}}$. Before message passing begins, a small gating network determines the relative importance of geometric and language (PLM) features for every residue. The gate first concatenates the two feature vectors, applies a linear projection, and normalizes the result with a softmax, $g_i = \text{Softmax}(\mathbf{W}_g[\mathbf{h}_i^{geo} \| \mathbf{h}_i^{plm}])$, where $\mathbf{W}_g \in \mathbb{R}^{2 \times d_h}$ is the weight matrix of the gate network with $d_h = d_{geo} + d_{plm}$. It then combines the inputs through feature-specific projections to the working width $d_h$:

$$\mathbf{h}_i^0 = \sum_{k \in \{geo, plm\}} g_{ik} \mathbf{W}_k \mathbf{h}_i^{(k)} \in \mathbb{R}^{d_h}.$$
(3)

The vector $\mathbf{h}_i^0$ serves as the initial node state for the first *EpiFormer* encoder block. The schematic of an *EpiFormer* block is shown in Figure 1 **(c)**. Let $\mathbf{H}_{ag}^\ell \in \mathbb{R}^{n \times d_h}$ and $\mathbf{H}_{ab}^\ell \in \mathbb{R}^{m \times d_h}$ be the current embeddings, which are passed in parallel to their EGNN-R and MHCA layers.

***Relation-aware EGNN (EGNN-R) layer***: We develop a relation-aware variant of EGNN (Satorras et al., 2021) to propagate structural and geometric information within each chain. Let $\mathbf{h}_i^\ell \in \mathbb{R}^{d_h}$ and $\mathbf{x}_i^\ell \in \mathbb{R}^3$ denote the feature and coordinate of residue $i$ after the $\ell$-th EGNN-R layer. Every undirected edge $e_{i,j}$ carries a tuple $\mathbf{r}_{i,j} \subseteq \mathcal{R}$ that encodes sequential and spatial relations. With the squared distance $d_{ij} = \|\mathbf{x}_i^\ell - \mathbf{x}_j^\ell\|_2^2$ and the displacement vector $\boldsymbol{\delta}_{ij} = \mathbf{x}_i^\ell - \mathbf{x}_j^\ell$, the layer performs the following computations:

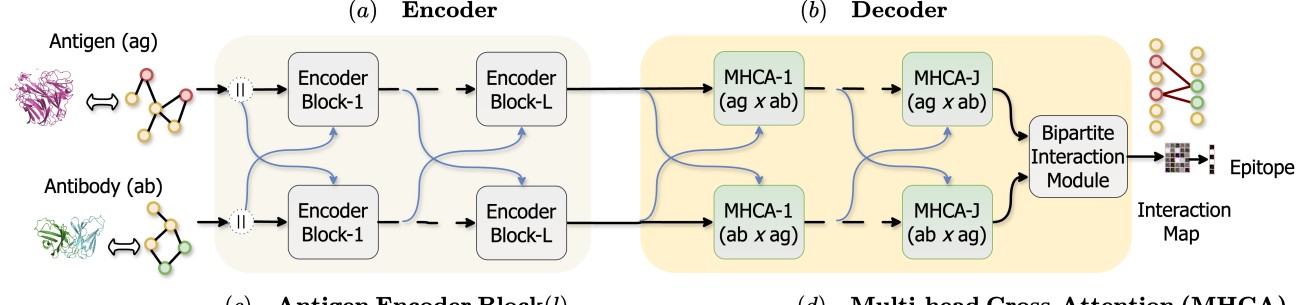

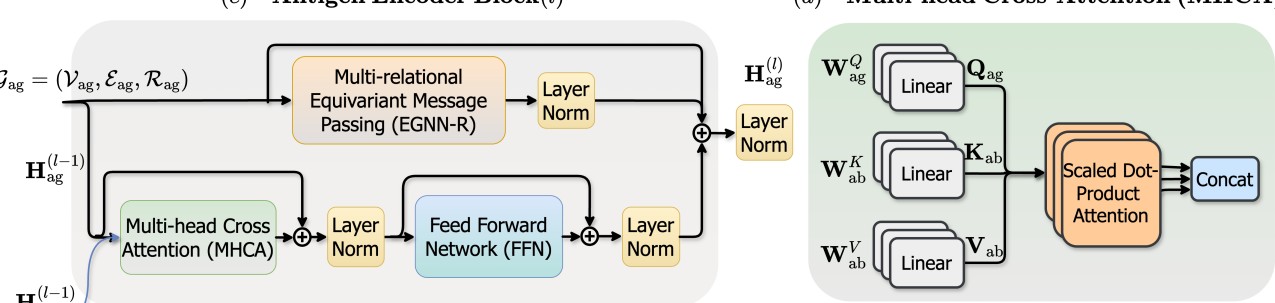

**Figure 1.** Overview of *EpiFormer*. The inputs are an antigen multi-relational graph $\mathcal{G}_{ag} = (\mathcal{V}_{ag}, \mathcal{E}_{ag}, \mathcal{R})$ and an antibody multi-relational graph $\mathcal{G}_{ab} = (\mathcal{V}_{ab}, \mathcal{E}_{ab}, \mathcal{R})$, while the outputs are the bipartite adjacency matrix and the binary epitope node labels. (a) Antigen and antibody graphs are encoded with parallel multi-relational equivariant message passing layers (EGNN-R) and cross-attention blocks. "||" is a small gating network that determines the relative importance of geometric and language features for every residue. (b) A bi-directional cross-attention decoder produces the interaction map. (c) Antigen Encoder Block schematic (Antibody Encoder Block is analogous) where "$\oplus$" denotes addition. (d) An example of MHCA between antigen and antibody.

$$m_{ij}^{\rho} = \phi_m^{\rho}\big(\mathbf{h}_i^{\ell}, \mathbf{h}_j^{\ell}, \gamma(d_{ij}), \mathbf{f}_{ij}\big), \qquad (4)$$

$$\mathbf{h}_i^{(\ell+1)} = \mathbf{h}_i^{\ell} + \phi_h\Big(\mathbf{h}_i^{\ell}, \sum_{j \in \mathcal{N}(i)} \sum_{\rho \in \mathbf{r}_{ij}} m_{ij}^{\rho}\Big), \qquad (5)$$

$$s_{ij}^{\rho} = \phi_x^{\rho}\big(m_{ij}^{\rho}\big), \qquad (6)$$

$$\mathbf{x}_i^{(\ell+1)} = \mathbf{x}_i^{\ell} + \sum_{j \in \mathcal{N}(i)} \sum_{\rho \in \mathbf{r}_{ij}} \frac{\boldsymbol{\delta}_{ij}}{\sqrt{d_{ij} + \varepsilon}}\, s_{ij}^{\rho}. \qquad (7)$$

Here, $\gamma(\cdot)$ denotes a 16-term radial basis function, $\mathbf{f}_{ij}$ is the edge's attribute vector, and each mapping $\phi_{\{m,x\}}^{\rho}$ is realized as a two-layer multilayer perceptron whose parameters are shared by all edges with the same relation label $\rho$, and $\varepsilon = 10^{-8}$. Specifically, we have four relation-specific message MLPs $\phi_m^{\rho} : \mathbb{R}^{2d_h + d_f + 16} \to \mathbb{R}^{d_q}$ and coordinate MLPs $\phi_x^{\rho} : \mathbb{R}^{d_q} \to \mathbb{R}^3$, and a node update MLP $\phi_h : \mathbb{R}^{d_h + d_q} \to \mathbb{R}^{d_h}$ shared across all relations, where $d_q$ represents hidden layer dimension. Applying residual connections and layer normalization produces output embeddings at layer $\ell$ as:

$$\mathbf{H}_{ag}^{intra} = \{\, W_{ag}^{\ell} \mathbf{h}_i^{\ell} \mid v_i \in \mathcal{V}_{ag} \,\}, \qquad (8)$$

$$\mathbf{H}_{ab}^{intra} = \{\, W_{ab}^{\ell} \mathbf{h}_j^{\ell} \mid v_j \in \mathcal{V}_{ab} \,\}, \qquad (9)$$

where $W^{\ell}$ represents the trainable parameters for the EGNN-R layer $\ell$ for each *EpiFormer* encoder block and $\mathbf{H}_{\{ag,ab\}}^{intra}$

represents the embeddings of antigen and antibody output residues after passing through their respective EGNN-R layer $\ell$. The layer remains E(3)-equivariant by construction because the only vector quantity entering the coordinate update is the displacement $\boldsymbol{\delta}_{ij}$, while cross-attention works with rotation and translation-invariant features (Liao & Smidt, 2022) (please refer to Appendix A.1 for the formal proof).

***Multi-head cross-attention layer with feed-forward network***:
In parallel to geometric message passing, each encoder block applies bidirectional multi-head cross-attention (MHCA) (Vaswani et al., 2017) to enable inter-chain communication. The MHCA mechanism shown in Figure 1 **(d)** produces cross-chain context representations $\widetilde{\mathbf{H}}_{ag}$ and $\widetilde{\mathbf{H}}_{ab}$. A learnable scalar gate $\alpha$ balances intra-chain geometry with cross-chain context:

$$\mathbf{H}_{ag}^{(\ell+1)} = \mathbf{H}_{ag}^{\ell} + \mathbf{H}_{ag}^{intra} + \alpha_{ag}\, \mathrm{FFN}(\widetilde{\mathbf{H}}_{ag}), \qquad (10)$$

$$\mathbf{H}_{ab}^{(\ell+1)} = \mathbf{H}_{ab}^{\ell} + \mathbf{H}_{ab}^{intra} + \alpha_{ab}\, \mathrm{FFN}(\widetilde{\mathbf{H}}_{ab}), \qquad (11)$$

where $\alpha_{ag}, \alpha_{ab} \in \mathbb{R}^+$ are learnable parameters, $\widetilde{\mathbf{H}} = \mathrm{MHCA}(\mathbf{H})$, and FFN is a two-layer Feed Forward Network. The MHCA is detailed in Appendix A.2.

**Decoder.** The decoder refines the residue embeddings $\mathbf{H}_{ag}^L$ and $\mathbf{H}_{ab}^L$ and performs bipartite interaction prediction.

The decoder has $J$ identical layers, each containing: (i) bidirectional MHCA with FFN, and (ii) layer normalization with residual connections, followed by a bipartite interaction head. The embeddings $\mathbf{H}_{\text{ag}}^J$ and $\mathbf{H}_{\text{ab}}^J$ serve as inputs to the bipartite interaction module.

***Bipartite interaction prediction module:*** The bipartite adjacency matrix is obtained by projecting the embeddings into queries and keys of width $d_k$ in both directions, forming scaled dot-product similarities:

$$\mathbf{S}_{\text{ag}\to\text{ab}} = \frac{(\mathbf{H}_{\text{ag}}^J \mathbf{W}_Q^{\text{out}})(\mathbf{H}_{\text{ab}}^J \mathbf{W}_K^{\text{out}})^\top}{\sqrt{d_k}}, \quad (12)$$

$$\mathbf{S}_{\text{ab}\to\text{ag}} = \frac{(\mathbf{H}_{\text{ab}}^J \mathbf{W}_Q'^{\text{out}})(\mathbf{H}_{\text{ag}}^J \mathbf{W}_K'^{\text{out}})^\top}{\sqrt{d_k}}. \quad (13)$$

The two score maps are fused via a learnable mixing vector $\mathbf{w} \in \mathbb{R}^2$ and bias $b \in \mathbb{R}$ to produce logits $\mathbf{Z} = \mathbf{w}^\top [\mathbf{S}_{\text{ag}\to\text{ab}} \ (\mathbf{S}_{\text{ab}\to\text{ag}})^\top] + b$, and the interaction probabilities are $\hat{\mathcal{E}}_{\text{bg}} = \sigma(\mathbf{Z}) \in \mathbb{R}^{n \times m}$.

### 3.3. Joint objective

*EpiFormer* is trained with a joint objective that consists of the primary epitope node classification loss and an auxiliary loss that includes a bipartite edge reconstruction and inter-chain geometric classification objective. The overall training objective is a weighted sum of these loss components:

$$\mathcal{L} = \lambda_{\text{node}} \mathcal{L}_{\text{node}} + \lambda_{\text{edge}} \mathcal{L}_{\text{edge}} + \lambda_{\text{geo}} \mathcal{L}_{\text{geo}}. \quad (14)$$

**Node Classification Loss ($\mathcal{L}_{\text{node}}$).** The node classification loss supervises epitope nodes only and combines three complementary objectives to handle class imbalance and enforce structural priors:

$$\mathcal{L}_{\text{node}} = \beta_{\text{BCE}} \mathcal{L}_{\text{BCE}}^{\text{epi}} + \beta_{\text{Dice}} \mathcal{L}_{\text{Dice}}^{\text{epi}} + \beta_{\text{sparsity}} \mathcal{L}_{\text{sparsity}}^{\text{epi}}, \quad (15)$$

where $\beta_{\{.\}}$ weight the different terms. The probability that node $v_{\text{ag}}$ is an epitope is derived from the bipartite interaction matrix using a top-$k$ pooling strategy which captures the relationship between $v_{\text{ag}}$ and nodes of the antibody:

$$(\hat{y}_{\text{ag}})_i = \frac{1}{k} \sum_{j \in \text{top-}k(\hat{\mathcal{E}}_{\text{bg}})_{i:}} (\hat{\mathcal{E}}_{\text{bg}})_{ij}, \quad (16)$$

where $(\hat{\mathcal{E}}_{\text{bg}})_{i:}$ denotes the $i$-th row, and $k$ is determined using cross-validation.

***Class-Reweighted Binary Cross-Entropy:*** The primary classification loss applies positive class reweighting ($\pi_{\text{epi}} > 1$) to address the severe class imbalance in epitope prediction:

$$\mathcal{L}_{\text{BCE}}^{\text{epi}} = -\frac{1}{n} \sum_{i=1}^{n} [\pi_{\text{epi}} (y_{\text{ag}})_i \log(\hat{y}_{\text{ag}})_i + (1 - (y_{\text{ag}})_i) \log(1 - (\hat{y}_{\text{ag}})_i)]. \quad (17)$$

***Dice Loss for Graph Segmentation:*** The Dice loss is effective for highly imbalanced image segmentation (Sudre et al., 2017) and treats epitope prediction as a segmentation problem:

$$\mathcal{L}_{\text{Dice}}^{\text{epi}} = 1 - \frac{2 \sum_{i=1}^{n} (\hat{y}_{\text{ag}})_i (y_{\text{ag}})_i + \alpha}{\sum_{i=1}^{n} (\hat{y}_{\text{ag}})_i + \sum_{i=1}^{n} (y_{\text{ag}})_i + \alpha}, \quad (18)$$

where $\alpha > 0$ is a small smoothing constant for numerical stability. The Dice coefficient measures the overlap between predicted and true epitope regions.

***Per-Graph Sparsity Regularization:*** The sparsity term enforces cardinality matching between predicted and true epitope counts for each complex in the mini-batch:

$$\mathcal{L}_{\text{sparsity}}^{\text{epi}} = ||\hat{y}_{\text{ag}} - y_{\text{ag}}||_1. \quad (19)$$

This regularizer is crucial for calibrating predictions across complexes of varying sizes.

**Edge Prediction Loss ($\mathcal{L}_{\text{edge}}$).** This loss applies positive-class-reweighted binary cross-entropy over all antigen-antibody residue pairs:

$$\mathcal{L}_{\text{edge}} = -\frac{1}{nm} \sum_{i=1}^{n} \sum_{j=1}^{m} [\pi_{\text{edge}} (\mathcal{E}_{\text{bg}})_{ij} \log(\hat{\mathcal{E}}_{\text{bg}})_{ij} + (1 - (\mathcal{E}_{\text{bg}})_{ij}) \log(1 - (\hat{\mathcal{E}}_{\text{bg}})_{ij})], \quad (20)$$

where $\mathcal{E}_{\text{bg}} \in \{0, 1\}^{n \times m}$ is the ground-truth interaction matrix per complex represented as a bipartite adjacency matrix $\hat{\mathcal{E}}_{\text{bg}}$ between antibody and antigen in the bipartite graph $\mathcal{G}_{\text{bg}} = (\mathcal{V}_{\text{ag}} \cup \mathcal{V}_{\text{ab}}, \mathcal{E}_{\text{bg}})$ and $\pi_{\text{edge}}$ compensates for the extreme sparsity of positives. An edge $e_{\text{bg}} \in \mathcal{E}_{\text{bg}}$ is a contact (labeled as 1) if the corresponding residues $(v_{\text{ag}}, v_{\text{ab}})$ are within 4.5Å of each other and 0 otherwise. This loss directly supervises the bipartite interaction prediction, which serves as the foundation for deriving epitope probabilities.

**Auxiliary Distance Classification Loss ($\mathcal{L}_{\text{geo}}$).** The auxiliary geometric term provides additional supervision by classifying inter-chain distances into discrete bins, helping the model learn geometrically meaningful representations. The loss focuses on near-contact pairs and ignores distant residue pairs that are unlikely to interact. This auxiliary supervision encourages the model to learn distance-aware representations while still maintaining focus on the primary epitope prediction task.

Let $\mathcal{M} = \{(i, j) : d_{ij} \leq D_{\max}\}$ be the set of antigen-antibody residue pairs within the maximum distance cutoff, where $d_{ij}$ is the Euclidean distance between residues $i$ and $j$. The bins are defined by distances $\{d_0, d_1, d_2, d_3, d_4\} = \{0, 4, 8, 16, 32\}$ Å, creating $B = 4$ bins:

$$b(i, j) = \arg \max_{b \in \{1, \ldots, 4\}} \mathbf{1}[d_{b-1} \leq d_{ij} < d_b]. \quad (21)$$

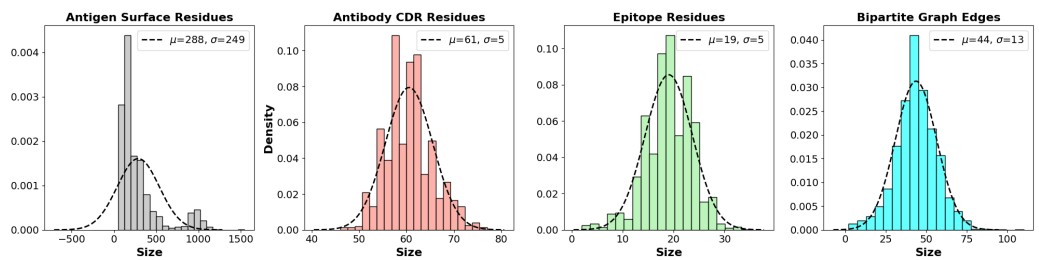

*Figure 2.* The size distribution of the antigen surface residues, antibody CDR residues, epitope residues, and the distribution of the number of edges in the antibody-antigen bipartite graph in the AsEP dataset.

The network predicts per-pair distance logits $\boldsymbol{\Delta}_{ij} \in \mathbb{R}^5$, but only the first $B = 4$ components $\widehat{\boldsymbol{\Delta}}_{ij} \in \mathbb{R}^4$ are used for pairs in $\mathcal{M}$, ignoring the "far" class beyond $D_{\max} = 32\,\text{Å}$. The class probabilities are:

$$p_{ijb} = \frac{\exp(\widehat{\Delta}_{ijb})}{\sum_{b'=1}^{4} \exp(\widehat{\Delta}_{ijb'})}. \tag{22}$$

The loss combines class balancing with distance weighting:

$$\mathcal{L}_{\text{geo}} = -\frac{1}{|\mathcal{M}|} \sum_{(i,j)\in\mathcal{M}} w_{ij} \sum_{b=1}^{4} \alpha_b \, \mathbf{1}\big[\,b(i,j) = b\,\big] \log p_{ijb}, \tag{23}$$

where $\alpha_b > 0$ are class-balance weights computed from empirical bin frequencies within $\mathcal{M}$ and $w_{ij} > 0$ are distance weights inversely proportional to $d_{ij}$, normalized to unit mean over $\mathcal{M}$.

## 4. Experiments

### 4.1. Settings

**Dataset.** We utilized the AsEP dataset (Liu et al., 2024), the largest benchmark of antibody-antigen complexes designed specifically for the epitope prediction task. After preprocessing, we retain 1,721 unique antibody–antigen complexes; details are in Appendix A.3.4. Figure 2 summarizes key dataset statistics.

*Stratified Splits:* We adopt two stratified splitting strategies from the AsEP benchmark dataset (Liu et al., 2024): **epitope-to-antigen surface ratio split** and **epitope-group split**. The first approach stratifies complexes by the ratio ($\#epitope\_nodes/\#antigen\_nodes$) to balance the class imbalance between interface and non-interface residues across train, validation, and test sets. Given that epitopes are typically limited in size (approximately $19 \pm 4.9$ residues), whereas antigen surfaces often contain several hundred residues, this stratification controls task difficulty by matching the distribution of epitope-to-surface ratios across splits.

The epitope-group split employs a different strategy by clustering complexes by antigen epitope and completely ex-

cluding test epitopes from training and validation data to evaluate model performance on novel binding sites. The dataset also includes multi-epitope antigens for which different antibodies bind distinct locations on the same antigen, and the split follows an 80/10/10 allocation by complexes. Both dataset splits result in 1,381 training complexes and 170 complexes each for validation and testing.

**Baseline Methods.** We trained several baseline methods for epitope prediction on the AsEP dataset using their reported training configurations. The baselines are broadly categorized into antigen / antibody-specific, protein-ligand binding site prediction, protein complex structure prediction, docking, binding affinity prediction, and molecular property prediction methods. Antigen / antibody-specific methods are the ones that are developed for the epitope prediction task, which include EpiGraph (Choi & Kim, 2024), EpiScan (Wang et al., 2024a), MIPE (Wang et al., 2024b), WALLE (Liu et al., 2024), DiscoTope3 (Høie et al., 2024), GraphBepi (Zeng et al., 2023), and PECAN (Pittala & Bailey-Kellogg, 2020). We also trained the current best protein-protein/ligand binding site prediction methods, such as EquiPocket (Zhang et al., 2023), AtomSurf (Mallet et al., 2023), and ESMBind (Schreiber, 2023), and evaluated their performance on the epitope prediction task. The protein complex structure prediction methods, such as Boltz-1 (Wohlwend et al., 2025) and AlphaFold-3 (Abramson et al., 2024), take the antigen-antibody sequence as input and predict the structure of the complex. The epitopic residues are then classified based on their proximity to the antibody in the predicted complex. We also re-trained binding affinity (CheapNet (Lim et al., 2025), Boltz-2 (Passaro et al., 2025), GearBind (Cai et al., 2024)) and molecular property prediction methods (Equiformer-v2 (Liao et al., 2023)) by adding classification heads to their last layer. To further test the robustness of EpiFormer, we compare its performance with a protein docking method, DiffDock (Corso et al., 2022). Table 1 presents the performance comparison of *EpiFormer* with baseline methods spanning these six categories, while the training details of these methods are provided in Appendix A.6. We evaluated the model performance using six standard classification metrics: Area Under the Receiver

*Table 1.* Performance comparison of baseline methods with *EpiFormer* (ours) on the AsEP dataset using the epitope-to-surface ratio and epitope-group stratified splits. The best values are represented in bold, while the second-best values are underlined.

| Method | Epitope-to-Surface Ratio Split | | | | | | Epitope Group Split | | | | | |
|---|---|---|---|---|---|---|---|---|---|---|---|---|
| | AUC↑ | AUPRC↑ | F1↑ | MCC↑ | Precision↑ | Recall↑ | AUC↑ | AUPRC↑ | F1↑ | MCC↑ | Precision↑ | Recall↑ |
| **Antigen / antibody-specific** | | | | | | | | | | | | |
| EpiGraph (Choi & Kim, 2024) | 0.794 | 0.227 | 0.078 | 0.103 | 0.355 | 0.044 | 0.779 | 0.194 | 0.056 | 0.096 | 0.401 | 0.030 |
| EpiScan (Wang et al., 2024a) | 0.669 | 0.135 | 0.203 | 0.124 | 0.148 | 0.327 | 0.443 | 0.080 | 0.127 | 0.048 | 0.089 | 0.221 |
| MIPE (Wang et al., 2024b) | 0.827 | 0.409 | 0.337 | 0.356 | **0.637** | 0.229 | 0.740 | **0.228** | 0.172 | 0.206 | **0.495** | 0.104 |
| WALLE (Liu et al., 2024) | 0.809 | 0.212 | 0.202 | 0.207 | 0.113 | **0.931** | 0.713 | 0.137 | 0.170 | 0.143 | 0.095 | **0.830** |
| DiscoTope3 (Høie et al., 2024) | 0.821 | 0.288 | 0.249 | 0.245 | 0.148 | 0.797 | 0.763 | 0.208 | 0.210 | 0.189 | 0.123 | 0.729 |
| GraphBepi (Zeng et al., 2023) | 0.783 | 0.232 | 0.079 | 0.118 | 0.430 | 0.044 | 0.781 | 0.220 | 0.144 | 0.161 | 0.386 | 0.088 |
| PECAN (Pittala & Bailey-Kellogg, 2020) | 0.740 | 0.156 | 0.229 | 0.182 | 0.150 | 0.488 | 0.729 | 0.166 | 0.254 | 0.195 | 0.169 | 0.513 |
| **Protein-ligand binding site prediction** | | | | | | | | | | | | |
| EquiPocket (Zhang et al., 2023) | 0.761 | 0.203 | 0.202 | 0.183 | 0.116 | 0.799 | 0.711 | 0.132 | 0.194 | 0.159 | 0.113 | 0.676 |
| AtomSurf (Mallet et al., 2023) | 0.606 | 0.094 | 0.139 | 0.062 | 0.076 | 0.817 | 0.699 | 0.144 | 0.202 | 0.151 | 0.127 | 0.500 |
| ESMBind (Schreiber, 2023) | 0.768 | 0.192 | 0.208 | 0.192 | 0.120 | 0.798 | 0.685 | 0.130 | 0.174 | 0.126 | 0.100 | 0.663 |
| **Protein structure prediction** | | | | | | | | | | | | |
| Boltz-1 (Wohlwend et al., 2025) | 0.771 | 0.242 | 0.257 | 0.223 | 0.155 | 0.750 | 0.702 | 0.174 | 0.229 | 0.169 | 0.146 | 0.536 |
| AlphaFold3 (Abramson et al., 2024) | 0.562 | 0.077 | 0.153 | 0.071 | 0.097 | 0.361 | 0.570 | 0.076 | 0.155 | 0.079 | 0.098 | 0.375 |
| **Binding affinity prediction** | | | | | | | | | | | | |
| CheapNet (Lim et al., 2025) | 0.731 | 0.244 | 0.178 | 0.147 | 0.100 | 0.818 | 0.641 | 0.095 | 0.152 | 0.100 | 0.084 | 0.776 |
| Boltz-2 (Passaro et al., 2025) | 0.789 | 0.277 | 0.254 | 0.227 | 0.151 | 0.794 | 0.702 | 0.168 | 0.220 | 0.156 | 0.140 | 0.518 |
| GearBind (Cai et al., 2024) | 0.799 | 0.215 | 0.244 | 0.236 | 0.144 | 0.787 | 0.734 | 0.154 | 0.206 | 0.169 | 0.123 | 0.627 |
| **Docking** | | | | | | | | | | | | |
| DiffDock (Corso et al., 2022) | 0.677 | 0.123 | 0.154 | 0.089 | 0.143 | 0.168 | 0.443 | 0.095 | 0.146 | 0.081 | 0.131 | 0.164 |
| **Molecular property prediction** | | | | | | | | | | | | |
| EquiFormer-v2 (Liao et al., 2023) | 0.831 | 0.279 | 0.255 | 0.260 | 0.150 | 0.846 | 0.729 | 0.160 | 0.212 | 0.169 | 0.131 | 0.561 |
| *EpiFormer* (ours) | **0.889** | **0.443** | **0.433** | **0.404** | 0.329 | 0.633 | **0.782** | 0.211 | **0.277** | **0.236** | 0.189 | 0.510 |

Operating Characteristic Curve (AUC-ROC), Area Under the Precision-Recall Curve (AUPRC), F1 score, Matthews Correlation Coefficient (MCC), precision, and recall.

## 4.2. Results.

*EpiFormer* achieves the best performance across the primary metrics on both evaluation splits. On the epitope-ratio split, *EpiFormer* obtains 0.889 AUC, 0.443 AUPRC, 0.433 F1, and 0.404 MCC. Compared to the next-best method MIPE (0.337 F1, 0.356 MCC), this corresponds to +28% relative improvement in F1 and +13% relative improvement in MCC. On the more challenging epitope-group split, where test epitopes are completely unseen during training, *EpiFormer* achieves 0.782 AUC and 0.277 F1, outperforming all baselines.

Antigen-specific methods (EpiGraph, GraphBepi, Disco-Tope3) achieve reasonable AUC but struggle with F1 and MCC due to ignoring antibody-specific binding patterns. Among antibody-aware methods, MIPE shows the strongest baseline performance. Methods from adjacent domains, such as Boltz-1, Boltz-2, GearBind, and Equiformer-v2, perform competitively but lack task-specific design for class-imbalanced epitope prediction. Notably, sophisticated docking methods like DiffDock (e.g., 0.154 F1) underperform simpler baselines, highlighting that architectural complexity alone does not guarantee strong epitope prediction.

*EpiFormer* optimizes the harmonic mean of precision (e.g., 0.329) and recall (e.g., 0.633), yielding superior F1 and MCC, which indicates a more discriminative model with a balanced trade-off. MIPE achieves higher precision by being overly conservative (0.637) but has a much lower recall (0.229), missing the majority of true epitopes. Methods optimized for recall predict most residues as epitopes (WALLE: 0.931), suffering from low precision with many false positives. We attribute the Dice loss and sparsity regularization in our joint objective to this balance.

**Qualitative Analysis.** Figure 3(a) visualizes the predicted interaction matrix for a representative complex (8DF5_3P). The ground truth shows a sparse binding interface concentrated in a specific antigen region. *EpiFormer*'s predictions closely match this pattern, where the model assigns high interaction scores to true binding residues (visible as a bright horizontal band) while correctly suppressing scores for non-epitope regions. This demonstrates that the model learns meaningful antigen-antibody interaction patterns rather than relying on surface-level features alone.

**Performance by Antigen Size.** Figure 3(b) shows that AU-ROC, MCC, and precision remain consistent across all size bins, indicating robust ranking ability. However, AUPRC, F1, and recall show a slightly decreasing trend as antigens grow larger, i.e., performance peaks for medium-sized antigens (100-200 residues) and drops for very large antigens (>500 residues). This trend reflects *EpiFormer*'s classification ability with increasing task difficulty, since epitopes constitute a smaller fraction of residues in larger antigens.

**Hyperparameter Sensitivity.** Figure 4 shows F1 sensitivity to seven loss hyperparameters. The BCE weight $\beta_{BCE}$ shows positive correlation with F1 score peaks around $\beta_{BCE} \approx 9$,

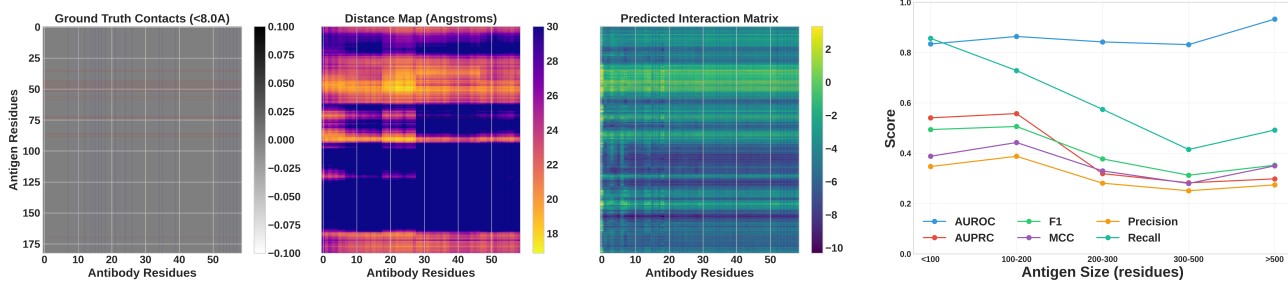

*Figure 3.* (a) Ground truth contacts, inter-chain distance map, and predicted interaction matrix by EpiFormer for complex 8DF5_3P. The binding interface is represented as a concentrated horizontal band corresponding to epitope residues. (b) Performance across antigen sizes.

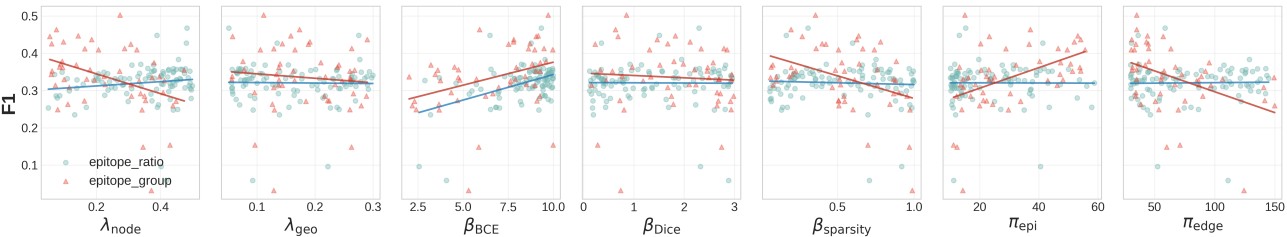

*Figure 4.* Sensitivity of F1 score to seven hyperparameters of the joint objective for epitope ratio and epitope group sweeping experiments.

while $\lambda_{node}$ and $\pi_{edge}$ show negative correlations favoring lower values for the epitope group setting. The remaining parameters ($\lambda_{geo}$, $\beta_{Dice}$, $\beta_{sparsity}$, $\pi_{epi}$) exhibit relatively flat trends, indicating robustness to their exact values.

## 4.3. Ablations

We conducted systematic ablations to quantify the contribution of each architectural component (full results with standard deviations in Appendix A.5). Our ablations reveal three design choices. First, **per-relation message passing** is essential: using multi-relational graphs with separate transformation functions per edge type yields 0.433 F1, compared to 0.337 for GearNet-style shared functions and 0.333 for simple proximity graphs (Table 2). This superiority shows that sequential edges and spatial edges require distinct learned transformations.

Second, **E(3)-equivariant encoding** substantially outperforms standard GNNs. EGNN-R achieves 0.433 F1 versus 0.337 for GCN, 0.326 for GAT, and 0.343 for REGNN (Wu et al., 2025) (Table 4). This gap between EGNN-R and REGNN isolates the contribution of geometric equivariance within the multi-relational framework.

Third, **cross-attention decoding** is important for capturing binding interactions. Cross-attention decoders achieve 0.433 F1 compared to 0.326 for dot-product alternatives, a 49% improvement (Table 5). The learnable gating parameters remain near their initialization at $\alpha \approx 0.05$, indicating that EGNN-R dominates the learned representations while cross-attention provides a small but consistent refinement

signal from the counterpart chain (Appendix Figure 6(b)).

Additional ablations show that top-2 pooling is optimal for aggregating interaction matrices (Table 6) and our joint loss formulation outperforms simpler objectives while contrastive learning degrades performance due to optimization conflicts (Table 7). Among the tested PLMs, ESM2-650M achieved the best results, outperforming both smaller and larger parameter variants (Table 3). We present the distinction of *EpiFormer* architecture and training objectives compared to the baseline methods in Table 8.

**Limitations.** While *EpiFormer* maintains strong AUROC across all antigen sizes, metrics like AUPRC show more variation for very large antigens (>500 residues), where class imbalance is most severe. Future work could explore SE(3)-equivariant alternatives (Fuchs et al., 2020), self-supervised pretraining (Zhang et al., 2022), and sequence-only variants using predicted structures. Testing the robustness of Epi-Former for general protein binding site prediction tasks is another promising direction for future work.

## 5. Conclusion

We presented *EpiFormer*, an encoder–decoder architecture for antibody-aware epitope prediction. Under comparable experimental conditions, *EpiFormer* outperforms prior methods on the AsEP benchmark dataset. Notably, our experiments suggest that coupling multi-relational geometric message passing with cross-attention at different levels is a promising direction for antibody-specific epitope prediction. Extensive ablations demonstrate the robustness of our work.

## Reproducibility Statement

We will make the code publicly available on GitHub and provide installation scripts to address the complex dependency issue of the libraries/packages. We hope that this will support and accelerate future research and development.

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

## A. Appendix

### A.1. E(3)-equivariance of the EGNN-R layer

**Theorem A.1** (E(3)-equivariance of the EGNN-R layer). *Consider the EGNN-R layer in §3.2 with updates*

$$m_{ij}^{\rho} = \phi_m^{\rho}\big(\mathbf{h}_i^{\ell}, \mathbf{h}_j^{\ell}, \gamma(d_{ij}), \mathbf{f}_{ij}\big), \tag{24}$$

$$s_{ij}^{\rho} = \phi_x^{\rho}\big(m_{ij}^{\rho}\big), \tag{25}$$

$$\mathbf{h}_i^{(\ell+1)} = \mathbf{h}_i^{\ell} + \phi_h\Big(\mathbf{h}_i^{\ell}, \sum_{j \in \mathcal{N}(i)} \sum_{\rho \in \mathbf{r}_{ij}} m_{ij}^{\rho}\Big), \tag{26}$$

$$\mathbf{x}_i^{(\ell+1)} = \mathbf{x}_i^{\ell} + \sum_{j \in \mathcal{N}(i)} \sum_{\rho \in \mathbf{r}_{ij}} \frac{\boldsymbol{\delta}_{ij}}{\sqrt{d_{ij} + \varepsilon}} s_{ij}^{\rho}, \tag{27}$$

*where $\boldsymbol{\delta}_{ij} = \mathbf{x}_i^{\ell} - \mathbf{x}_j^{\ell}$, $d_{ij} = \|\boldsymbol{\delta}_{ij}\|_2^2$, and $\varepsilon > 0$. Assume: (i) node features $\mathbf{h}_i^{\ell} \in \mathbb{R}^{d_h}$ are scalar channels, (ii) $\mathbf{h}_{ij}$ and $\mathbf{r}_{ij}$ are categorical and independent of coordinates, (iii) $\gamma$ is any scalar function of $d_{ij}$, (iv) each $\phi_{\{m,x\}}^{\rho}$ is an MLP from scalars to scalars. Let the $E(3)$ action be $g = (R, t)$ with $R \in O(3)$ and $t \in \mathbb{R}^3$, acting as $\mathbf{x}_i^{\ell} \mapsto R\mathbf{x}_i^{\ell} + t$ and $\mathbf{h}_i^{\ell} \mapsto \mathbf{h}_i^{\ell}$. Then the layer is $E(3)$-equivariant:*

$$\big\{ \mathbf{x}_i^{\ell}, \mathbf{h}_i^{\ell} \big\}_{i=1}^n \mapsto \big\{ R\mathbf{x}_i^{\ell} + t, \mathbf{h}_i^{\ell} \big\}_{i=1}^n \implies \big\{ \mathbf{x}_i^{(\ell+1)}, \mathbf{h}_i^{(\ell+1)} \big\}_{i=1}^n \mapsto \big\{ R\mathbf{x}_i^{(\ell+1)} + t, \mathbf{h}_i^{(\ell+1)} \big\}_{i=1}^n.$$

*Consequently, any stack of such layers is $E(3)$-equivariant by composition.*

*Proof.* Let $g = (R, t) \in E(3)$ act as stated. Edge data $\mathbf{f}_{ij}$ and $\mathbf{r}_{ij}$ are unchanged.

*Invariants.* Relative displacement and distance transform as

$$\boldsymbol{\delta}_{ij} \mapsto R\boldsymbol{\delta}_{ij}, \qquad d_{ij} = \|\boldsymbol{\delta}_{ij}\|^2 \mapsto \|R\boldsymbol{\delta}_{ij}\|^2 = d_{ij}. \tag{28}$$

Hence $d_{ij}$, $\gamma(d_{ij})$, and $(d_{ij} + \varepsilon)^{-1/2}$ are invariant scalars.

*Scalar messages and coefficients.* Each message $m_{ij}^{\rho} = \phi_m^{\rho}(\mathbf{h}_i^{\ell}, \mathbf{h}_j^{\ell}, \gamma(d_{ij}), \mathbf{f}_{ij})$ depends only on scalars that are invariant under $g$, so $m_{ij}^{\rho}$ is invariant. Then $s_{ij}^{\rho} = \phi_x^{\rho}(m_{ij}^{\rho})$ is also invariant.

*Feature update.* The update

$$\mathbf{h}_i^{(\ell+1)} = \mathbf{h}_i^{\ell} + \phi_h\Big(\mathbf{h}_i^{\ell}, \sum_{j \in \mathcal{N}(i)} \sum_{\rho \in \mathbf{r}_{ij}} m_{ij}^{\rho}\Big) \tag{29}$$

uses only invariant scalars, so $\mathbf{h}_i^{(\ell+1)}$ is invariant. This matches the scalar action on features.

*Coordinate update.* The increment

$$\Delta\mathbf{x}_i = \sum_{j \in \mathcal{N}(i)} \sum_{\rho \in \mathbf{r}_{ij}} \frac{\boldsymbol{\delta}_{ij}}{\sqrt{d_{ij} + \varepsilon}} s_{ij}^{\rho} \tag{30}$$

is a sum of relative vectors scaled by invariant scalars. Under $g$ each term becomes

$$\frac{\boldsymbol{\delta}_{ij}}{\sqrt{d_{ij} + \varepsilon}} s_{ij}^{\rho} \mapsto \frac{R\boldsymbol{\delta}_{ij}}{\sqrt{d_{ij} + \varepsilon}} s_{ij}^{\rho} = R\Big(\frac{\boldsymbol{\delta}_{ij}}{\sqrt{d_{ij} + \varepsilon}} s_{ij}^{\rho}\Big), \tag{31}$$

so $\Delta\mathbf{x}_i \mapsto R\,\Delta\mathbf{x}_i$. Therefore

$$\mathbf{x}_i^{(\ell+1)} = \mathbf{x}_i^{\ell} + \Delta\mathbf{x}_i \mapsto R\mathbf{x}_i^{\ell} + t + R\Delta\mathbf{x}_i = R\big(\mathbf{x}_i^{\ell} + \Delta\mathbf{x}_i\big) + t = R\mathbf{x}_i^{(\ell+1)} + t. \tag{32}$$

*Composition.* The composition of equivariant maps is equivariant. Hence, any stack of EGNN-R layers is $E(3)$-equivariant. $\square$

**A.2. Multi-head cross-attention with feed-forward network (MHCA)**

The bidirectional multi-head cross-attention mechanism enables information exchange between antigen and antibody chains. Let $n_{\text{head}}$ be the number of heads with per-head width $d_a = d_h/n_{\text{head}}$. For layer $\ell$, independent linear projections produce queries, keys, and values:

$$\mathbf{Q}_{\text{ag}}^{\ell} = \mathbf{H}_{\text{ag}}^{(\ell-1)}\mathbf{W}_{\text{ag}}^{Q(\ell)}, \tag{33}$$

$$\mathbf{K}_{\text{ab}}^{\ell} = \mathbf{H}_{\text{ab}}^{(\ell-1)}\mathbf{W}_{\text{ab}}^{K(\ell)}, \tag{34}$$

$$\mathbf{V}_{\text{ab}}^{\ell} = \mathbf{H}_{\text{ab}}^{(\ell-1)}\mathbf{W}_{\text{ab}}^{V(\ell)}, \tag{35}$$

with analogous expressions for the reverse direction. After reshaping to $n_{\text{head}}$ heads of width $d_a$, scaled dot-product attention computes the affinity matrices:

$$\mathbf{A}_{\text{ag}\leftarrow\text{ab}}^{\ell} = \text{softmax}\Big(\tfrac{1}{\sqrt{d_h}}\,\mathbf{Q}_{\text{ag}}^{\ell}{\mathbf{K}_{\text{ab}}^{\ell}}^{\top} + \mathbf{M}\Big), \tag{36}$$

where $\mathbf{M}$ is a batch mask (applied only in decoder) that assigns $-\infty$ to residue pairs from different complexes. The resulting context vectors are:

$$\widetilde{\mathbf{H}}_{\text{ag}}^{\ell} = \big[\mathbf{A}_{\text{ag}\leftarrow\text{ab}}^{\ell}\mathbf{V}_{\text{ab}}^{\ell}\big]\mathbf{W}_{O,\text{ag}}^{\ell}, \tag{37}$$

$$\widetilde{\mathbf{H}}_{\text{ab}}^{\ell} = \big[\mathbf{A}_{\text{ab}\leftarrow\text{ag}}^{\ell}\mathbf{V}_{\text{ag}}^{\ell}\big]\mathbf{W}_{O,\text{ab}}^{\ell}. \tag{38}$$

Each direction then applies a feed-forward network $\text{FFN}(\mathbf{x}) = \mathbf{W}_2\,\sigma(\mathbf{W}_1\mathbf{x} + \mathbf{b}_1) + \mathbf{b}_2$ with dropout, residual connections, and layer normalization.

**A.3. Graph Construction**

A.3.1. NODE FEATURES

Each residue node in our protein graph incorporates two complementary information sources that together provide a rich representation of both local structural properties and evolutionary context:

**Local geometry & physicochemistry.** Each residue $v_i \in \mathcal{V}$ is annotated with a 105-dimensional geometric and biochemical feature vector $\mathbf{h}_i^{\text{geo}} \in \mathbb{R}^{d_{\text{geo}}}$ that encodes the type, position, distance, direction, angle, and orientation of each residue. Such residue-level descriptors are widely employed in diverse protein-related studies in structural bioinformatics (Wu et al., 2025; Jing et al., 2020; Jumper et al., 2021). This vector is constructed as follows:

$$\mathbf{h}_i^{\text{geo}} = \Big[E_{\text{type}}(v_i),\ E_{\text{pos}}(i),\ \sin(\eta_i),\ \cos(\eta_i),\ \text{RBF}(\|\mathbf{x}_{i,C_\alpha} - \mathbf{x}_{i,\xi}\|),\ Q_i^{\top}\frac{\mathbf{x}_{i,\xi} - \mathbf{x}_{i,C_\alpha}}{\|\mathbf{x}_{i,\xi} - \mathbf{x}_{i,C_\alpha}\|}\Big], \tag{39}$$

where:

- $E_{\text{type}}$: Embedding for amino acid residue type (e.g., arginine, glycine).

- $E_{\text{pos}}$: Positional encoding of residue index in the sequence, enabling the model to distinguish between identical amino acids based on their sequence context. This positional information is crucial for understanding long-range dependencies and structural motifs, as amino acids at different sequence positions (N-terminus vs. C-terminus, loop regions vs. secondary structures) often play different functional roles even if they are the same amino acid type.

- $\eta_i$: Local backbone geometry encoded through six fundamental angles that determine how the protein chain folds at each residue $v_i$ and are encoded by their sine and cosine (12 scalars). Bond angles $(\alpha_i, \beta_i, \gamma_i)$ describe the geometric constraints of covalent bonds, while dihedral angles $(\psi_i, \phi_i, \omega_i)$ capture the rotational freedom that gives rise to secondary structures like helices and sheets.

- $\text{RBF}(\cdot)$: Radial basis function encoding distances between $C_\alpha$ and other backbone atoms ($\xi \in \{C_\beta, N, O\}$), with each distance represented by 16 Gaussian basis functions.

---

**Algorithm 1** *EpiFormer*: High-Level Architecture

---

**Input:** Antigen graph $\mathcal{G}_{ag}$ and antibody graph $\mathcal{G}_{ab}$ with coordinates $\mathbf{X}$, features $\mathbf{h}^{geo}$, $\mathbf{h}^{plm}$

**Output:** Bipartite interaction matrix $\hat{\mathcal{E}}_{bg} \in [0,1]^{n \times m}$

// **Feature Initialization**

**1 foreach** *chain* $\in \{ag, ab\}$ **do**

**2** $\quad$ Apply gating network to combine geometric and PLM features $\mathbf{h}_i^0 \leftarrow \text{Gate}(\mathbf{h}_i^{geo}, \mathbf{h}_i^{plm})$ for each residue $i$

**3 end**

// **Encoder: Parallel Processing**

**4 for** *layer* $\ell = 1$ **to** $L$ **do**

**5** $\quad$ // Intra-chain geometric message passing

$\quad (\mathbf{H}_{ag}^{intra}, \mathbf{X}^{ag}) \leftarrow \text{EGNN-R}(\mathcal{G}_{ag}, \mathbf{H}_{ag}^{(\ell-1)}, \mathbf{X}^{ag})$ $(\mathbf{H}_{ab}^{intra}, \mathbf{X}^{ab}) \leftarrow \text{EGNN-R}(\mathcal{G}_{ab}, \mathbf{H}_{ab}^{(\ell-1)}, \mathbf{X}^{ab})$

$\quad$ // Inter-chain cross-attention

**6** $\quad \widetilde{\mathbf{H}}_{ag} \leftarrow \text{MHCA}(\mathbf{H}_{ag}^{intra}, \mathbf{H}_{ab}^{intra}, \mathbf{H}_{ag}^{intra})$ $\widetilde{\mathbf{H}}_{ab} \leftarrow \text{MHCA}(\mathbf{H}_{ab}^{intra}, \mathbf{H}_{ag}^{intra}, \mathbf{H}_{ab}^{intra})$

$\quad$ // Combine intra-chain and cross-chain information

**7** $\quad \mathbf{H}_{ag}^{\ell} \leftarrow \mathbf{H}_{ag}^{(\ell-1)} + \mathbf{H}_{ag}^{intra} + \alpha_{ag} \text{FFN}(\widetilde{\mathbf{H}}_{ag})$ $\mathbf{H}_{ab}^{\ell} \leftarrow \mathbf{H}_{ab}^{(\ell-1)} + \mathbf{H}_{ab}^{intra} + \alpha_{ab} \text{FFN}(\widetilde{\mathbf{H}}_{ab})$

**8 end**

// **Decoder: Cross-Attention Refinement**

**9** Initialize decoder embeddings: $\mathbf{H}_{ag}^{dec} \leftarrow \mathbf{H}_{ag}^L$, $\mathbf{H}_{ab}^{dec} \leftarrow \mathbf{H}_{ab}^L$ **for** *layer* $\ell = 1$ **to** $L$ **do**

$\quad$ // Inter-chain cross-attention

**10** $\quad \widetilde{\mathbf{H}}_{ag}^{dec} \leftarrow \text{MHCA}(\mathbf{H}_{ag}^{dec}, \mathbf{H}_{ab}^{dec}, \mathbf{H}_{ab}^{dec})$ $\widetilde{\mathbf{H}}_{ab}^{dec} \leftarrow \text{MHCA}(\mathbf{H}_{ab}^{dec}, \mathbf{H}_{ag}^{dec}, \mathbf{H}_{ag}^{dec})$

$\quad$ // Combine intra-chain and cross-chain information

**11** $\quad \mathbf{H}_{ag}^{dec(\ell)} \leftarrow \mathbf{H}_{ag}^{dec(\ell-1)} + \text{FFN}(\widetilde{\mathbf{H}}_{ag}^{dec})$ $\mathbf{H}_{ab}^{dec(\ell)} \leftarrow \mathbf{H}_{ab}^{dec(\ell-1)} + \text{FFN}(\widetilde{\mathbf{H}}_{ab}^{dec})$

**12 end**

// **Bipartite Interaction Prediction**

**13** Compute bidirectional attention scores: $\mathbf{S}_{ag \to ab} \leftarrow \frac{(\mathbf{H}_{ag}^{dec} \mathbf{W}_Q^{out})(\mathbf{H}_{ab}^{dec} \mathbf{W}_K^{out})^\top}{\sqrt{d_k}}$ $\mathbf{S}_{ab \to ag} \leftarrow \frac{(\mathbf{H}_{ab}^{dec} \mathbf{W}_Q'^{out})(\mathbf{H}_{ag}^{dec} \mathbf{W}_K'^{out})^\top}{\sqrt{d_k}}$

**14** Fuse scores and apply sigmoid: $\mathbf{Z} \leftarrow \mathbf{w}^\top [\mathbf{S}_{ag \to ab} (\mathbf{S}_{ab \to ag})^\top] + b$ $\hat{\mathcal{E}}_{bg} \leftarrow \sigma(\mathbf{Z})$

// **Epitope Extraction**

**15** Extract per-residue epitope probabilities via top-$k$ pooling: $(\hat{y}_{ag})_i = \frac{1}{k} \sum_{j \in \text{top-}k(\hat{\mathcal{E}}_{bg})_{i:}} (\hat{\mathcal{E}}_{bg})_{ij}$

**16 Function** MHCA($Q, K, V$)**:**

**17** $\quad Q_h \leftarrow QW_Q^h, K_h \leftarrow KW_K^h, V_h \leftarrow VW_V^h$ ; $\qquad\qquad$ // Project per head $h$

**18** $\quad \alpha_{ij}^h \leftarrow \text{Softmax}_j \left( \frac{Q_{h,i} \cdot K_{h,j}^\top}{\sqrt{d_h}} \right)$ ; $\qquad\qquad$ // Attention scores

**19** $\quad C_i^h \leftarrow \sum_j \alpha_{ij}^h V_{h,j}$ ; $\qquad\qquad$ // Context vector

$\quad$ **Result:** $\text{Concat}(C^1, \dots, C^H)W_O$

$\quad$ ; $\qquad\qquad$ // Combine heads

**20 end**

**21 Function** FFN($X$)**:**

**22** $\quad \hat{\mathcal{E}}_{bg} \leftarrow \text{SiLU}(XW_1 + b_1)W_2 + b_2$ ; $\qquad\qquad$ // $W_1 \in \mathbb{R}^{d \times d_{ff}}$, $W_2 \in \mathbb{R}^{d_{ff} \times d}$

$\quad$ **Result:** $\hat{\mathcal{E}}_{bg}$

**23 end**

---

- $Q_i^\top \mathbf{u}_i$: Here, $Q_i \in \mathbb{R}^{3 \times 3}$ is the orthonormal rotation matrix defining the local coordinate system constructed from the $\mathbf{C}_\alpha$, $\mathbf{C}_\beta$, and N atoms of residue $i$, and $\mathbf{u}_i = [\mathbf{u}_i^1, \mathbf{u}_i^2, \mathbf{u}_i^3] \in \mathbb{R}^{3 \times 3}$ contains the normalized direction vectors between these atoms (e.g., $\mathbf{u}_i^1 = \frac{\mathbf{x}_{i,C_\beta} - \mathbf{x}_{i,C_\alpha}}{\|\mathbf{x}_{i,C_\beta} - \mathbf{x}_{i,C_\alpha}\|}$). The matrix product $Q_i^\top \mathbf{u}_i$ transforms these direction vectors into the local coordinate frame and is flattened to yield a 9-dimensional feature vector. Note that the oxygen atom is stored in the coordinate matrix for other calculations (like the RBF distance features), but isn't used for the local coordinate frame

construction.

- The coordinates are held in a $3 \times 4$ matrix, which is used in the calculation of node and edge features.

$$\mathbf{X}_i = \begin{bmatrix} \mathbf{x}_{i,\mathrm{N}} & \mathbf{x}_{i,\mathrm{C}_\alpha} & \mathbf{x}_{i,\mathrm{C}_\beta} & \mathbf{x}_{i,\mathrm{O}} \end{bmatrix} \quad \in \mathbb{R}^{3\times4}, \quad \text{where} \quad \mathbf{x}_{i,\xi} \in \mathbb{R}^3$$

**Frozen protein language model (PLM) embeddings.** We extract embeddings for the antigen and antibody sequences $\mathbf{z}_i^{\mathrm{plm}} \in \mathbb{R}^{d_c}$ using pre-trained protein-language models (e.g., ESM-2 (Lin et al., 2023)) to provide the model an orthogonal information source (evolutionary + biochemical context). Since the original PLM embeddings are high-dimensional (for example, $d_c = 1280$ for ESM2-650M), we project them to a lower-dimensional representation suitable for our architecture:

$$\mathbf{h}_i^{\mathrm{plm}} = \mathbf{W}_{\mathrm{plm}} \mathbf{z}_i^{\mathrm{plm}}, \quad \text{where} \quad \mathbf{W}_{\mathrm{plm}} \in \mathbb{R}^{d_{\mathrm{plm}} \times d_c}. \tag{40}$$

Here, $d_{\mathrm{plm}}$ is the target dimensionality for the compressed PLM features, and $\mathbf{W}_{\mathrm{plm}}$ serves as a learnable bottleneck that adapts the frozen PLM representations to our specific task.

### A.3.2. EDGE FEATURES

We compute a 100-dimensional edge feature vector $\mathbf{f}_{i,j} \in \mathbb{R}^{d_f}$ that describes the spatial and sequential relationship between two residues $v_i$ and $v_j$. This vector integrates multiple complementary descriptors to provide a rich representation of inter-residue interactions (Jing et al., 2020) and is defined as follows:

$$\mathbf{f}_{i,j} = \left\{ E_{\mathrm{type}}(e_{i,j}), E_{\mathrm{pos}}(i-j), \mathrm{RBF}(\|\mathbf{x}_{i,\mathrm{C}_\alpha} - \mathbf{x}_{j,\xi}\|), Q_i^\top \frac{\mathbf{x}_{j,\xi} - \mathbf{x}_{i,\mathrm{C}_\alpha}}{\|\mathbf{x}_{j,\xi} - \mathbf{x}_{i,\mathrm{C}_\alpha}\|}, q\left(Q_i^\top Q_j\right) \mid \xi \right\}, \tag{41}$$

where $E_{\mathrm{type}}(e_{i,j})$ is the one-hot encoding of relations $\mathbf{r}_{i,j}$ of length 4 between two residues, and the positional encoding $E_{\mathrm{pos}}(i-j)$ encodes the relative sequential position sinusoidally to 16 scalars. The third and fourth terms are distance and direction encodings of four backbone atoms $\xi$ in residue $v_j$ in the local coordinate frame $Q_i$. These four inter-residue distances $\{d(\mathrm{C}_\alpha, \mathrm{C}_\beta), d(\mathrm{C}_\alpha, \mathrm{N}), d(\mathrm{C}_\alpha, \mathrm{O}), d(\mathrm{C}_\alpha, \mathrm{C}_\alpha)\}$ are each represented by 16 Gaussian basis functions. The last term $q\left(Q_i^\top Q_j\right)$ is the quaternion representation $q(\cdot)$ of $Q_i^\top Q_j$. By integrating sequence position, local geometry, and orientation, the model understands the residue identity from global pose and enables robust generalization across structures. These node and edge features are visualized in Figure 5(a).

### A.3.3. EDGE RELATIONS

Since spatial proximity between residues alone cannot capture hydrogen bonding's directional specificity or electrostatic complementarity's charge-based selectivity, we use multi-relational edges to capture distinct interaction types (Zhang et al., 2022). By treating each relation separately, the model learns complex interaction patterns within the protein. Hence, to expand the contexts of these interactions, we divide the edges into four different types of relations $\mathcal{R} = \{\rho_1, \rho_2, \rho_3, \rho_4\}$, including *(i)* **sequential relations** $\rho_1$ and $\rho_2$ between two residues with relative sequential distance equal to 1 (peptide bond) and 2 (short-range torsion coupling); *(ii)* **spatial relations** between residues that are from the same component and spatially connected due to $K$-nearest neighbors (relation $\rho_3$ that captures local packing shell) or with a Euclidean distance less than 8Å (relation $\rho_4$) capturing medium-range contact between residues within the protein structure (Wu et al., 2025).

To illustrate the importance of edge relations, consider a discontinuous epitope spanning two antigen loops: sequential edges $(\rho_1, \rho_2)$ maintain the structural integrity of each loop, while spatial edges $(\rho_3, \rho_4)$ capture the three-dimensional proximity between residues from different loops, enabling the model to understand how distant sequence regions come together to form a cohesive binding interface. We provide a schematic of edge relations in Fig. 5 (b), where each edge $e_{i,j} \in \mathcal{E}$ is associated with a set of relations $\mathbf{r}_{i,j} \in \mathcal{R}$. Besides, two relations $\rho_1$ (with sequence distance equal to 1) can derive a relation $\rho_2$ (with sequence distance equal to 2), while an edge may connect two nodes (residues) due to both relations $\rho_3$ and $\rho_4$.

### A.3.4. PREPROCESSING

For each complex, we first separated the paired antigen and antibody chains into individual structure files. We then performed sequence-structure alignment using Clustal Omega (Sievers et al., 2011) to establish correspondence between SEQRES (complete sequence) and ATOMSEQ (resolved atoms) records. This alignment generated binary masks that enable reliable mapping of sequences to structural residues (seqres2surf and seqres2cdr) while preserving the native crystallographic ordering.

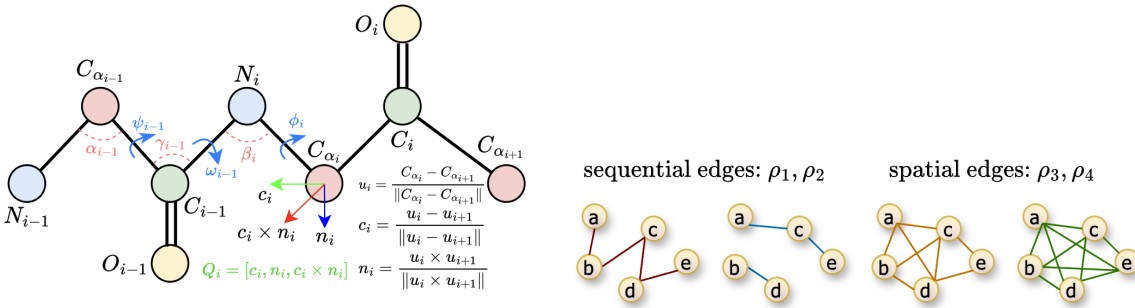

*Figure 5.* (a) Node and edge features encoding position, distance, direction, angle, and orientation (Figure credit: (Wu et al., 2025)). (b) Four edge relations (sequential $\rho_1$, $\rho_2$; spatial $\rho_3$, $\rho_4$). To avoid complexity, we visualize only some edges.

For antibody chains, we applied the alignment masks to reindex heavy (H) and light (L) chains by removing insertion codes to enforce consecutive 1-based residue numbering required for graph construction. Antigen chains underwent similar processing to maintain parity between sequences and structures. This step ensures that each residue in the protein sequence corresponds exactly to its structural counterpart during the graph representation. Then, we applied solvent-accessibility filters to retain only antigen surface residues, using the original AsEP seqres2surf masks to define the node set for antigen residue graphs. The binary epitope labels were projected onto the surface ATOMSEQ via alignment masks, while paratope labels were preserved for antibody residue nodes. This surface filtering step prevents non-surface residues from confounding epitope supervision while maintaining all necessary information for cross-chain interaction modeling.

We excluded two complexes (5nj6_0P and 5ies_0P) from the AsEP dataset due to sequence alignment inconsistencies and unresolved residues, with the final dataset containing 1,721 complexes. The contact distribution between residues in the bipartite graph had a mean of 43.7 contacts with a standard deviation of 12.8. Additionally, the dataset includes 641 unique antigens and 973 epitope groups, highlighting the diversity and complexity of the antibody-antigen interactions captured in the AsEP dataset.

To incorporate evolutionary and semantic information, we integrated embeddings from state-of-the-art PLMs. For antigens, we extracted embeddings using the ESM model family, while for antibodies, we incorporated AntiBERTy embeddings (Ruffolo et al., 2023; Ahmed et al., 2025), a transformer model specialized for antibody sequences, providing better functional and evolutionary context for paratope regions. These embeddings were mapped to graph nodes using the seqres2atmseq alignment masks. Finally, we used these preprocessed structures to generate HeteroData objects for the multi-relational graphs using PyTorch Geometric (Fey & Lenssen, 2019).

### A.4. Implementation details

The model is trained with an Adam optimizer and a ReduceLROnPlateau learning-rate schedule with decoupled weight decay. The learning rate is selected from the sweep-defined range and fixed at approximately $6.5e{-}5$ in the best configuration. A ReduceLROnPlateau scheduler monitors validation performance and decays the learning rate on stagnation, while an early stopping with patience of 10 epochs prevents overfitting and reduces variance in final selection. We used SiLU activation functions (Elfwing et al., 2018) throughout the model because they provide stable gradients via their smooth, non-monotonic curve, which are crucial for training deep graph networks. The hyperparameter tuning was performed via a Bayesian optimization sweep in Weights & Biases to minimize the validation loss, and the best hyperparameters were chosen within a predefined search space using bounded uniform and log-uniform distributions. We ran the hyperparameter sweeping experiments separately for the epitope ratio and epitope group split settings.

- The model weight decay was sampled log-uniformly over $[1e{-}6,\ 1e{-}4]$ to prevent overfitting by penalizing large weights, with the best configuration using approximately $9.9e{-}5$.

- The model dropout was sampled log-uniformly over $[0.05, 0.5]$ to improve the generalizability of the model, and the best performing configuration for the epitope ratio model used a dropout of 0.132, while the epitope group model used a dropout of 0.053.

- The number of layers in the encoder module is treated as a hyperparameter and was chosen from the set $[3, 4, 5]$ while

for the decoder, the number of layers was chosen from the $[2, 3, 4]$. We experimented with different encoder hidden dimensions and the best configuration of 128 was picked from $[64, 128, 256, 512]$ across different runs.

- We also experimented with different number of attention heads for the encoder and decoder MHCA (2,4,8,16) and picked the best model with 8 attention heads.

- A batch size of 8 was chosen from [4,8,16,32] across different runs.

- $\alpha_{ag}$ and $\alpha_{ab}$ are initialised to 0.05

For the loss coefficients, the best for the model trained on the epitope group split run uses $\lambda_{edge} = 1.0$, $\lambda_{node} = 0.4816$, $\lambda_{geo} = 0.0514$, $\beta_{BCE} = 9.3249$, $\beta_{Dice} = 2.2966$, $\beta_{sparsity} = 0.3068$, $\pi_{epi} = 15.2856$, $\pi_{edge} = 58.7077$, label smoothing $\epsilon = 0.1$, and a distance cutoff of 32 Å for $\mathcal{L}_{geo}$. While, the best run for the model trained on the epitope group split uses $\lambda_{node} = 0.143$, $\lambda_{edge} = 1.0$, $\lambda_{geo} = 0.158$, $\beta_{BCE} = 9.16$, $\beta_{Dice} = 1.83$, $\beta_{sparsity} = 0.64$, $\pi_{epi} = 53.18$, $\pi_{edge} = 44.11$, label smoothing $\epsilon = 0.1$, and a distance cutoff of 32 Å for $\mathcal{L}_{geo}$.

- The bipartite edge positive-class weight $\pi_{edge}$ for the BCE-with-logits interaction loss was sampled log-uniformly over $[30, 150]$, accommodating variation in pairwise sparsity across complexes.

- The node objective weight $\lambda_{node}$ was sampled uniformly over $[0.05, 0.5]$, exploring the trade-off between residue supervision and the other objectives.

- The binary cross-entropy multiplier within the node objective $\beta_{BCE}$ was drawn uniformly over $[2, 10]$, spanning weak to strong emphasis on classification error.

- The Dice multiplier $\beta_{Dice}$ was drawn uniformly over $[0.1, 3.0]$, reflecting its role as a secondary calibrator under class imbalance.

- The epitope positive-class weight $\pi_{epi}$ was sampled log-uniformly over $[10, 60]$, covering roughly an order of magnitude in imbalance without biasing toward either extreme.

- The per-graph epitope count-regularizer weight $\beta_{sparsity}$ was sampled uniformly over $[0.05, 1.0]$, enabling calibration of predicted positive counts at the complex level.

- The auxiliary distance-classification weight $\lambda_{geo}$ was sampled uniformly over $[0.05, 0.3]$, with class balancing across distance bins and distance-aware pair weighting kept enabled and the maximum distance fixed at 32 Å for all trials.

The experiments were performed on an NVIDIA RTX 6000 GPU and it took around 35-60 minutes for a single hyperparameter sweeping experiment of around 50 epochs. To ensure full reproducibility of our experiments, we implement random seed management across all computational components, including NumPy (`numpy.random`), Python (`random`), PyTorch (`torch.manual_seed`), and CUDA operations (`torch.cuda.manual_seed_all`), while additionally controlling worker initialization in data loaders and disabling non-deterministic algorithms (`torch.backends.cudnn.deterministic=True`).

## A.5. Ablation Studies

We performed ablation studies on the different protein graph representations, model components such as encoder and decoder architectures, pooling strategies, and loss functions. Unless otherwise noted, the results are reported as mean $\pm$ standard deviation over 3 random seeds.

### A.5.1. GRAPH CONSTRUCTION

This ablation isolates how residue-level graph design affects *EpiFormer*'s antibody-specific epitope prediction by holding node/edge features, PLM inputs, and training configuration fixed while swapping the underlying graph topology. Specifically, we compared three protein graph representations: a simple residue-only graph that collapses relations into proximity edges (Choi & Kim, 2024), a RAAD-style multi-relational graph with four edge types (sequential and spatial) (Wu et al., 2025), and a GearNet (Zhang et al., 2022) variant with seven relation types constructed to capture finer-grained structural

neighborhoods. The node and edge features were fixed for all three graph types, and the edge relations were only varied. This design quantifies the contribution of relation granularity and edge semantics of the proteins to the downstream performance of epitope prediction. Table 2 compares the epitope prediction performance of *EpiFormer* using the three graph representations.

*Table 2.* Performance metrics for different protein graph representation architectures on epitope prediction tasks. All values are reported for the epitope-to-surface ratio split. The best values are represented in bold, while the second-best values are underlined.

| Graph type | AUC | AUPRC | F1 | MCC | Precision | Recall |
|---|---|---|---|---|---|---|
| Simple | 0.821 | 0.355 | 0.333 | 0.294 | 0.240 | 0.543 |
| GearNet | 0.812 | 0.315 | 0.337 | 0.286 | 0.290 | 0.401 |
| Multi-relational | **0.889** | **0.443** | **0.433** | **0.404** | **0.329** | **0.633** |

We also performed experiments by using different sequence embeddings from the Evolutionary Scale Modeling (ESM) family to explore their contribution to the epitope prediction task. We used three variants of the ESM2 (Lin et al., 2023) model family (35M, 650M, and 3B parameters) as well as the newer ESM3-small (Hayes et al., 2025) model (1.4B parameters). Our experiments in Table 3 show that ESM2-650M produces the best contextual features for the antigen-antibody binding site prediction task. We attribute the important role of PLMs in handling the data scarcity issue for the antibody-aware epitope prediction task.

*Table 3.* Performance metrics for different PLM embeddings on the epitope prediction tasks. 4 models from the Evolutionary Scale Modeling (ESM) family were used to generate embeddings for antigens, while AntiBERTy (IgFold) was used to generate embeddings for the antibodies. All values are reported for the epitope-to-surface ratio split. The best values are represented in bold, while the second-best values are underlined.

| PLM | AUC | AUPRC | F1 | MCC | Precision | Recall |
|---|---|---|---|---|---|---|
| ESM2-35M | 0.815 | 0.330 | 0.334 | 0.283 | 0.287 | 0.399 |
| ESM2-650M | **0.889** | **0.443** | **0.433** | **0.404** | **0.329** | **0.633** |
| ESM2-3B | 0.826 | 0.331 | 0.349 | 0.300 | 0.285 | 0.449 |
| ESM3-small | 0.840 | 0.374 | 0.377 | 0.330 | 0.331 | 0.437 |

### A.5.2. GNNs

To assess the impact of geometric message passing on epitope prediction performance, we systematically replaced the EGNN-R layers in the encoder of *EpiFormer* with alternative GNN architectures. We evaluated standard GNN variants, including graph convolutional network (GCN) (Kipf, 2016), graph isomorphism network (GIN) (Xu et al., 2018), graph attention transformer (GAT) (Veličković et al., 2017), as well as more sophisticated approaches such as relational graph convolutional network (RGCN) (Schlichtkrull et al., 2018), and relation-aware equivariant graph network (REGNN) (Wu et al., 2025), as shown in Table 4. Traditional GNNs like GCN, GIN, and GAT perform competitively but below EGNN-R, highlighting the importance of incorporating geometric equivariance for accurate modeling of three-dimensional protein binding interfaces. EGNN-R achieves 0.433 F1 compared to 0.337 for GCN, 0.326 for GAT, and 0.343 for REGNN, representing a 42% improvement over the next-best equivariant baseline.

*Table 4.* Performance comparison of different GNNs used in the *EpiFormer* encoder blocks on epitope prediction tasks. The best values are represented in bold, while the second-best values are underlined.

| Model | AUC | AUPRC | F1 | MCC | Precision | Recall |
|---|---|---|---|---|---|---|
| EGNN-R | **0.889 $\pm$ 0.045** | **0.443 $\pm$ 0.130** | **0.433 $\pm$ 0.014** | **0.404 $\pm$ 0.235** | **0.329 $\pm$ 0.067** | **0.633 $\pm$ 0.030** |
| GAT | 0.827 $\pm$ 0.006 | 0.308 $\pm$ 0.021 | 0.326 $\pm$ 0.010 | 0.276 $\pm$ 0.012 | 0.263 $\pm$ 0.016 | 0.435 $\pm$ 0.062 |
| GCN | 0.831 $\pm$ 0.006 | 0.325 $\pm$ 0.009 | 0.337 $\pm$ 0.010 | 0.290 $\pm$ 0.010 | 0.264 $\pm$ 0.014 | 0.467 $\pm$ 0.016 |
| GIN | 0.826 $\pm$ 0.007 | 0.310 $\pm$ 0.022 | 0.333 $\pm$ 0.016 | 0.284 $\pm$ 0.019 | 0.270 $\pm$ 0.004 | 0.437 $\pm$ 0.043 |
| REGNN | 0.833 $\pm$ 0.005 | 0.334 $\pm$ 0.015 | 0.343 $\pm$ 0.015 | 0.294 $\pm$ 0.015 | 0.276 $\pm$ 0.025 | 0.453 $\pm$ 0.016 |
| RGCN | 0.824 $\pm$ 0.004 | 0.314 $\pm$ 0.016 | 0.325 $\pm$ 0.008 | 0.276 $\pm$ 0.009 | 0.255 $\pm$ 0.018 | 0.452 $\pm$ 0.042 |

### A.5.3. MODEL

We also replaced the cross-attention decoder with dot-product and dual alternatives. The **dot-product** decoder computes the interaction matrix as a plain inner product between antigen and antibody embeddings and produces a fast and parameter-free similarity score. The **dual** decoder architecture integrates two parallel processing paths: a dot-product similarity route

and a sparse cross-attention mechanism, and merges their outputs via a learnable weight $\alpha$. The ablation studies show that cross-attention decoders substantially outperform alternatives, achieving 0.433 F1 compared to 0.326 for dot-product and 0.334 for dual decoders. This 49% improvement over dot-product decoding demonstrates the importance of learned cross-chain attention for capturing binding interactions. Dot-product decoding produces lower recall (0.464 vs 0.633), whereas cross-attention preserves a stronger precision-recall balance as shown in Table 5.

Figure 7 visualizes the learned representations via t-SNE projections, showing clear separation between epitope and non-epitope residues. We further tested the class separability of *EpiFormer* by computing the silhouette scores, which demonstrate a progressive improvement from early encoder layers through the decoder, as shown in Figure 6(a).

*Table 5.* Performance comparison of different decoder blocks for epitope prediction. The best values are represented in bold, while the second-best values are underlined.

| Decoder | AUC | AUPRC | F1 | MCC | Precision | Recall |
|---|---|---|---|---|---|---|
| Cross Attn. | **0.889 ± 0.045** | **0.443 ± 0.130** | **0.433 ± 0.014** | **0.404 ± 0.235** | **0.329 ± 0.067** | **0.633 ± 0.030** |
| Dot Product | 0.827 ± 0.009 | 0.315 ± 0.034 | 0.326 ± 0.011 | 0.278 ± 0.015 | 0.252 ± 0.009 | 0.464 ± 0.053 |
| Dual | 0.834 ± 0.008 | 0.329 ± 0.030 | 0.334 ± 0.014 | 0.286 ± 0.017 | 0.266 ± 0.008 | 0.450 ± 0.033 |

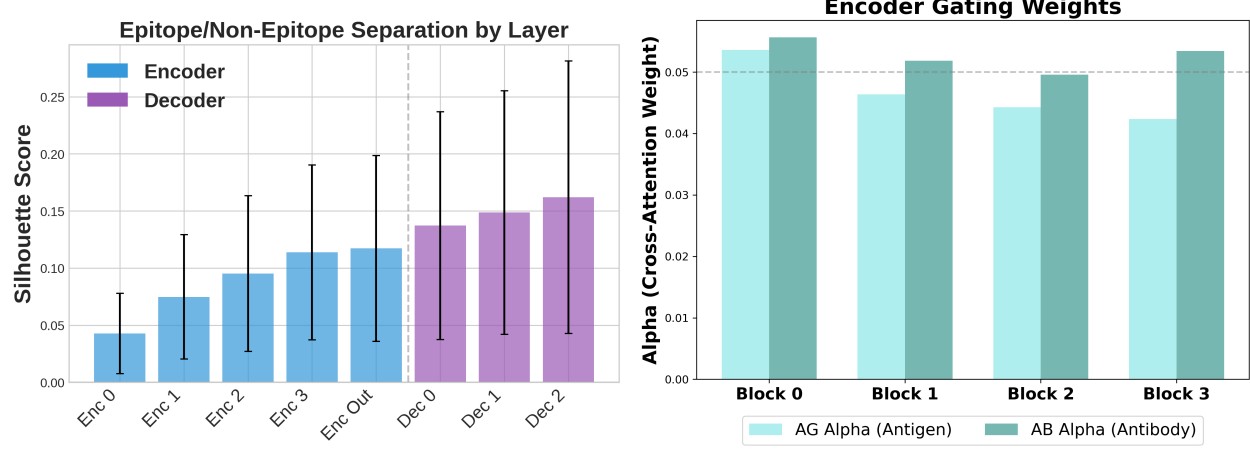

*Figure 6.* (a) Layer-wise silhouette scores measuring class separability, demonstrating progressive refinement from encoder through decoder layers. (b) The gating weights ($\alpha_{\text{ag}}$ and $\alpha_{\text{ab}}$) across the encoder layers show how the model learned to balance between local geometric message passing (EGNN-R) and global cross-chain attention for each encoder block.

We performed ablation studies over different pooling strategies. We map the bipartite interaction matrix $\hat{\mathcal{E}}_{\text{bg}}$ to per-residue probabilities by aggregating across the partner dimension (row-wise for epitopes, column-wise for paratopes): **Max pooling** assigns the maximum interaction per residue; **Mean pooling** averages interactions over all partners; **Top-$k$ mean pooling** averages the largest $k$ interactions (small $k$, e.g., 2) to reflect a few key partners; **Softmax-attention** converts interactions to attention weights via a softmax along the partner dimension and returns the weighted sum; **Hierarchical pooling** takes a convex combination of top-2 mean (local specificity) and global mean (context) with a mixing weight $\alpha$. Empirically (Table 6), Top-2 pooling yields the highest AUC/AUPRC/F1, hierarchical pooling is competitive, while max/mean/softmax-attention and larger $k$ underperform and tend to over-concentrate probability mass and impair calibration.

*Table 6.* Performance comparison of different pooling methods for epitope prediction. The best values are represented in bold, while the second-best values are underlined.

| Pooling Method | AUC | AUPRC | F1 | MCC | Precision | Recall |
|---|---|---|---|---|---|---|
| Hierarchical Pooling | 0.836 ± 0.004 | 0.338 ± 0.012 | 0.341 ± 0.009 | 0.295 ± 0.006 | 0.268 ± 0.022 | 0.476 ± 0.038 |
| Max | 0.830 ± 0.005 | 0.321 ± 0.021 | 0.326 ± 0.006 | 0.279 ± 0.011 | 0.265 ± 0.029 | 0.441 ± 0.085 |
| Mean | 0.834 ± 0.006 | 0.324 ± 0.016 | 0.332 ± 0.004 | 0.283 ± 0.004 | 0.281 ± 0.024 | 0.414 ± 0.048 |
| Pool Top-2 | **0.889 ± 0.045** | **0.443 ± 0.130** | **0.433 ± 0.014** | **0.404 ± 0.235** | **0.329 ± 0.067** | **0.633 ± 0.030** |
| Pool Top-3 | 0.851 ± 0.030 | 0.370 ± 0.062 | 0.370 ± 0.048 | 0.330 ± 0.059 | 0.286 ± 0.034 | 0.529 ± 0.103 |
| Pool Top-4 | 0.836 ± 0.008 | 0.342 ± 0.019 | 0.340 ± 0.018 | 0.295 ± 0.019 | 0.260 ± 0.020 | 0.493 ± 0.020 |
| Softmax Attn. | 0.832 ± 0.007 | 0.329 ± 0.018 | 0.332 ± 0.004 | 0.285 ± 0.005 | 0.256 ± 0.006 | 0.472 ± 0.020 |

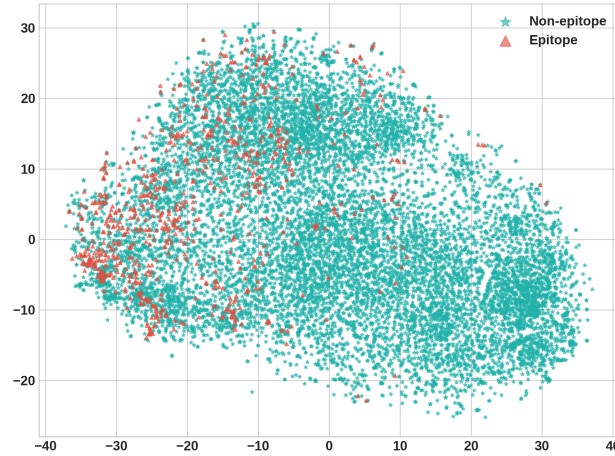

*Figure 7.* t-SNE visualization of antigen residue embeddings colored by epitope (red) and non-epitope (blue) labels showing learned class separation.

### A.5.4. LOSS

We performed ablations to evaluate the contribution of the loss function/s (primary, auxiliary, and regularizers) on the epitope prediction task, as shown in Table 7.

**Contrastive Learning Loss ($\mathcal{L}_{\textbf{InfoNCE}}$).**  We also performed contrastive learning with the SimCLR InfoNCE (Information Noise Contrastive Estimation) loss (Chen et al., 2020) to learn discriminative representations by contrasting positive and negative residue pairs within and across protein chains. The contrastive loss combines intra-chain and inter-chain objectives:

$$\mathcal{L}_{\text{contrastive}} = \lambda_{\text{intra}}\mathcal{L}_{\text{intra}} + \lambda_{\text{inter}}\mathcal{L}_{\text{inter}}, \tag{42}$$

where $\lambda_{\text{intra}}$ and $\lambda_{\text{inter}}$ balance the relative importance of within-chain and cross-chain contrastive learning.

**Intra-Chain Contrastive Loss ($\mathcal{L}_{\textbf{intra}}$).**  The intra-chain loss encourages similar representations for residues with the same label (epitope/non-epitope or paratope/non-paratope) within each protein chain:

$$\mathcal{L}_{\text{intra}} = \mathcal{L}_{\text{intra}}^{\text{ag}} + \mathcal{L}_{\text{intra}}^{\text{ab}}. \tag{43}$$

For each chain (antigen or antibody), the loss is computed as:

$$\mathcal{L}_{\text{intra}}^{\text{chain}} = -\frac{1}{|\mathcal{P}|} \sum_{i \in \mathcal{P}} \log \frac{\sum_{j \in \mathcal{P}_{i+}} \exp(\mathbf{h}_i^T \mathbf{h}_j / \tau)}{\sum_{k \in \mathcal{N}_i} \exp(\mathbf{h}_i^T \mathbf{h}_k / \tau)}, \tag{44}$$

where $\mathcal{P} = \{i : y_i = 1\}$ is the set of positive (binding) residues, $\mathcal{P}_{i+} = \{j \in \mathcal{P} : j \neq i\}$ are other positive residues sharing the same label as anchor $i$, $\mathcal{N}_i$ includes all negative residues for anchor $i$, $\mathbf{h}_i, \mathbf{h}_j$ are $L_2$-normalized residue embeddings, and $\tau$ is the temperature parameter controlling concentration.

**Inter-Chain Contrastive Loss ($\mathcal{L}_{\textbf{inter}}$).**  The inter-chain loss promotes alignment between epitope and paratope representations across antigen-antibody pairs:

$$\mathcal{L}_{\text{inter}} = \mathcal{L}_{\text{ag}\rightarrow\text{ab}} + \mathcal{L}_{\text{ab}\rightarrow\text{ag}}. \tag{45}$$

The bidirectional formulation ensures symmetric learning:

$$\mathcal{L}_{\text{ag}\rightarrow\text{ab}} = -\frac{1}{|\mathcal{P}_{\text{ag}}|} \sum_{i \in \mathcal{P}_{\text{ag}}} \log \frac{\sum_{j \in \mathcal{P}_{\text{ab}}} \exp(\mathbf{h}_i^{\text{ag}T} \mathbf{h}_j^{\text{ab}} / \tau)}{\sum_{k \in \mathcal{N}_{\text{cross}}} \exp(\mathbf{h}_i^{\text{ag}T} \mathbf{h}_k / \tau)}, \tag{46}$$

where $\mathcal{P}_{\text{ag}}, \mathcal{P}_{\text{ab}}$ are epitope and paratope residue sets, $\mathcal{N}_{\text{cross}}$ includes negative residues from both chains, and the loss pulls epitope embeddings closer to paratope embeddings while pushing them away from non-binding residues. Our experiments

show that contrastive learning didn't contribute to improving the classification performance. We attribute this to conflicting optimization objectives between BCE loss and standard InfoNCE loss, a phenomenon demonstrated in a recent work (Ji et al., 2024).

*Table 7.* Performance comparison of different loss function configurations for epitope prediction. All metrics are reported for epitope prediction tasks. The best values are represented in bold, while the second-best values are underlined.

| Loss Configuration | AUC | AUPRC | F1 | MCC | Precision | Recall |
|---|---|---|---|---|---|---|
| $\mathcal{L}_{\mathrm{bce}}$ | $0.822 \pm 0.013$ | $0.274 \pm 0.038$ | $0.199 \pm 0.017$ | $0.205 \pm 0.022$ | $0.111 \pm 0.011$ | $0.946 \pm 0.016$ |
| $\mathcal{L}_{\mathrm{edge}}$ | $0.581 \pm 0.006$ | $0.098 \pm 0.002$ | $0.142 \pm 0.010$ | $0.086 \pm 0.007$ | $0.154 \pm 0.002$ | $0.132 \pm 0.016$ |
| $\mathcal{L}_{\mathrm{bce}} + \mathcal{L}_{\mathrm{geo}}$ | $0.822 \pm 0.009$ | $0.266 \pm 0.025$ | $0.205 \pm 0.012$ | $0.214 \pm 0.013$ | $0.115 \pm 0.008$ | $0.941 \pm 0.019$ |
| $\mathcal{L}_{\mathrm{bce}} + \mathcal{L}_{\mathrm{edge}}$ | $0.826 \pm 0.006$ | $0.296 \pm 0.018$ | $0.220 \pm 0.008$ | $0.230 \pm 0.009$ | $0.125 \pm 0.005$ | $0.914 \pm 0.011$ |
| $\mathcal{L}_{\mathrm{bce}} + \mathcal{L}_{\mathrm{edge}} + \mathcal{L}_{\mathrm{dice}}$ | $0.818 \pm 0.011$ | $0.268 \pm 0.029$ | $0.205 \pm 0.017$ | $0.210 \pm 0.021$ | $0.115 \pm 0.011$ | $0.930 \pm 0.026$ |
| $\mathcal{L}_{\mathrm{bce}} + \mathcal{L}_{\mathrm{edge}} + \mathcal{L}_{\mathrm{geo}}$ | $0.826 \pm 0.015$ | $0.299 \pm 0.050$ | $0.214 \pm 0.020$ | $0.223 \pm 0.022$ | $0.121 \pm 0.013$ | $0.926 \pm 0.033$ |
| $\mathcal{L}_{\mathrm{edge}} + \mathcal{L}_{\mathrm{node}} + \mathcal{L}_{\mathrm{geo}}$ | $\mathbf{0.889 \pm 0.045}$ | $\mathbf{0.443 \pm 0.130}$ | $\mathbf{0.433 \pm 0.014}$ | $\mathbf{0.404 \pm 0.235}$ | $\mathbf{0.329 \pm 0.067}$ | $\mathbf{0.633 \pm 0.030}$ |
| $\mathcal{L}_{\mathrm{edge}} + \mathcal{L}_{\mathrm{node}} + \mathcal{L}_{\mathrm{geo}} + \mathcal{L}_{\mathrm{InfoNCE}}$ | $0.850 \pm 0.031$ | $0.362 \pm 0.064$ | $0.361 \pm 0.051$ | $0.323 \pm 0.064$ | $0.270 \pm 0.034$ | $0.550 \pm 0.111$ |
| $\mathcal{L}_{\mathrm{bce}} + \mathcal{L}_{\mathrm{edge}} + \mathcal{L}_{\mathrm{InfoNCE}}$ | $0.837 \pm 0.002$ | $0.345 \pm 0.007$ | $0.338 \pm 0.008$ | $0.296 \pm 0.005$ | $0.254 \pm 0.020$ | $0.511 \pm 0.049$ |
| $\mathcal{L}_{\mathrm{bce}} + \mathcal{L}_{\mathrm{edge}} + \mathcal{L}_{\mathrm{sparsity}}$ | $0.835 \pm 0.002$ | $0.336 \pm 0.013$ | $0.334 \pm 0.006$ | $0.288 \pm 0.007$ | $0.270 \pm 0.026$ | $0.453 \pm 0.078$ |
| $\mathcal{L}_{\mathrm{edge}} + \mathcal{L}_{\mathrm{node}}$ | $0.835 \pm 0.003$ | $0.325 \pm 0.012$ | $0.329 \pm 0.001$ | $0.283 \pm 0.004$ | $0.260 \pm 0.026$ | $0.462 \pm 0.071$ |
| $\mathcal{L}_{\mathrm{edge}} + \mathcal{L}_{\mathrm{node}} + \mathcal{L}_{\mathrm{InfoNCE}}$ | $0.829 \pm 0.006$ | $0.305 \pm 0.015$ | $0.326 \pm 0.010$ | $0.276 \pm 0.012$ | $0.261 \pm 0.006$ | $0.435 \pm 0.023$ |

## A.6. Baseline Comparison

We report the distinction of *EpiFormer* in terms of its architecture and training objectives with comparison to the baseline methods as shown in Table 8.

*Table 8.* Comprehensive comparison of baseline methods and *EpiFormer*. **Architecture**: Ab (antibody-aware), St (structure), PLM (pretrained embeddings), Graph (GNN), Geom (geometric features), Multi-rel (multi-relational edges), E(3) (equivariant), X-Att (cross-attention), Inter (interleaved attention), Gate (learnable gating). **Loss**: BCE (binary cross-entropy), Dice (Dice loss), Focal (focal loss), Contr (contrastive), Count (count regularizer), Edge (edge prediction loss).

| Method | Ab | St | PLM | Graph | Architecture Geom | M-rel | E(3) | X-Att | Inter | Gate | BCE | Dice | Loss Components Focal | Contr | Count | Edge |
|---|---|---|---|---|---|---|---|---|---|---|---|---|---|---|---|---|
| EpiGraph† | ✗ | ✓ | ✓ | ✓ | ✗ | ✗ | ✗ | ✗ | ✗ | ✗ | ✓ | ✗ | ✗ | ✗ | ✗ | ✗ |
| GraphBepi† | ✗ | ✓ | ✓ | ✓ | ✓ | ✗ | ✗ | ✗ | ✗ | ✗ | ✓ | ✗ | ✗ | ✗ | ✗ | ✗ |
| EquiPocket† | ✗ | ✓ | ✗ | ✓ | ✓ | ✗ | ✓ | ✗ | ✗ | ✗ | ✓ | ✓ | ✗ | ✗ | ✗ | ✗ |
| EpiScan | ✓ | ✗ | ✓ | ✗ | ✗ | ✗ | ✗ | ✗ | ✗ | ✗ | ✓ | ✗ | ✗ | ✗ | ✗ | ✗ |
| WALLE | ✓ | ✓ | ✓ | ✓ | ✗ | ✗ | ✗ | ✗ | ✗ | ✗ | ✓ | ✗ | ✗ | ✗ | ✓ | ✓ |
| PECAN | ✓ | ✓ | ✗ | ✓ | ✓ | ✗ | ✗ | ✓ | ✗ | ✗ | ✓ | ✗ | ✗ | ✗ | ✗ | ✗ |
| MIPE | ✓ | ✓ | ✓ | ✓ | ✓ | ✓ | ✗ | ✓ | ✗ | ✗ | ✓ | ✗ | ✗ | ✓ | ✗ | ✗ |
| *EpiFormer* | ✓ | ✓ | ✓ | ✓ | ✓ | ✓ | ✓ | ✓ | ✓ | ✓ | ✓ | ✓ | ✓ | ✓ | ✓ | ✓ |

†Antigen-only (no antibody information).

All baseline methods were evaluated on the AsEP benchmark dataset consisting of 1,721 antibody-antigen complexes (after excluding problematic samples 5nj6_0P and 5ies_0P). To ensure fair comparison, all methods used identical experimental settings: official AsEP paper splits (supporting both epitope-ratio stratified split and epitope-group clustered split), 5-fold cross-validation, random seed 42, classification threshold 0.5, and consistent evaluation metrics (AUROC, AUPRC, F1, MCC, precision, recall). Below, we describe the architecture and adaptation details for each baseline.

A.6.1. PROTEIN / PROTEIN-LIGAND BINDING SITE PREDICTION METHODS

**EquiPocket (Zhang et al., 2023)** operates at atom-level with 6D features, computing molecular surfaces via MSMS with 7D local geometric descriptors. The Surface-EGNN module performs E(3)-equivariant multi-channel convolution with DenseNet-style connections. We adapted to epitope prediction by adding residue aggregation (CA atom selection or mean pooling) to convert per-atom outputs to per-residue predictions.

**AtomSurf (Mallet et al., 2023)** jointly encodes molecular surfaces using DiffusionNet (spectral convolution with 22D geometric features including HKS) and residue graphs using GCN with ProNet features, connected through bidirectional KNN-based message passing. We preserve the joint surface-graph encoding for antigen-only processing with mean-pooled surface features aggregated to the residue level.

**ESMBind (Schreiber, 2023)** adapts the ESM-2 protein language model (35M parameters by default) with LoRA (Low-Rank Adaptation) for parameter-efficient fine-tuning, training only 0̃.5% of parameters. The model processes antigen amino acid sequences through ESM-2 with LoRA adapters (rank $r = 4$, scaling $\alpha = 8$, dropout 0.1) applied to attention layers, followed by a per-residue classification head for epitope prediction. We fine-tune ESMBind on the epitope sequences using class-weighted BCE loss to handle label imbalance, and support multiple ESM-2 variants (8M to 650M parameters).

### A.6.2. Protein Structure Prediction Methods

**Boltz-1 (Wohlwend et al., 2025)** uses a Pairformer trunk with triangle multiplication ($z_{ij}+ = \sum_k a_{ik} \cdot b_{jk}$), triangle attention, and attention with pair bias for structure-aware representations. We adapt by projecting ESM-2 embeddings to single (384D) and pairwise (128D) representations, reducing to 8 layers, and adding an epitope prediction head on concatenated single and mean-pooled pairwise features.

**AlphaFold3 (Abramson et al., 2024)** uses a diffusion-based architecture with Pairformer trunk processing single and pairwise representations through adaptive LayerNorm, gated linear units, and triangle operations. We use inference-only mode with pretrained weights [1], converting inter-chain contact predictions from predicted complexes to epitope labels using distance thresholds. Due to imprecise docking without evolutionary information, we used a relaxed 15Å contact threshold. AlphaFold3 without MSA achieved AUC=0.56 with the relaxed 15Å threshold, only marginally better than random, confirming that structure prediction alone without co-evolutionary signals cannot reliably identify epitopes.

### A.6.3. Docking Methods

**DiffDock (Corso et al., 2022)** generates ligand poses through diffusion over SE(3) using E(3)-equivariant score networks. We use inference-only mode with DiffDock-L to dock antibody CDR regions onto antigen structures, labeling antigen residues within contact distance of predicted CDR poses as epitopes, weighted by confidence scores.

### A.6.4. Binding Affinity Prediction Methods

**CheapNet (Lim et al., 2025)** uses Geometry-Informed Graph Neural Network blocks for local structure encoding with intra/inter-molecular edges and cross-attention between entity representations. We adapted from graph-level affinity regression to node-level classification by replacing ligand-protein atom pairs with antibody-antigen residue pairs, projecting 35D residue features from RAAD (Wu et al., 2025)/ESM-2, and adding a per-residue MLP classifier with BCE loss.

**GearBind (Cai et al., 2024)** is a pretrainable geometric GNN based on GearNet, pretrained on CATH using contrastive learning and fine-tuned on SKEMPI for $\Delta\Delta G_{\text{bind}}$ prediction. We extract encoder representations for antigen residues, replace the affinity regression head with a binary classifier, and initialize from CATH-pretrained weights.

**Boltz-2 (Passaro et al., 2025)** extends Boltz-1 with cross-attention between binding partners. We additionally process antibody embeddings (AntiBERTy, 512D) alongside antigen (ESM-2, 1280D), using 2-layer CrossPairAttention for antibody-aware epitope prediction with the same triangle-based Pairformer backbone.

### A.6.5. Molecular Property Prediction Methods

**EquiformerV2 (Liao et al., 2023)** is an improved equivariant transformer using SO(2) convolutions for efficiency, processing 3D graphs with type-$L$ features and attention in irreducible representation space. We adapt from energy/force regression to residue-level classification by constructing antigen graphs with CA coordinates, combining RAAD features with ESM-2 embeddings, and replacing the output head with per-residue binary classification.

---

[1]The AF3 model weights were obtained from DeepMind. We used AF3 without MSA due to computational resource constraints.

