# OpenReview forum: "EpiFormer: Learning Antigen-Antibody Interactions for Epitope Prediction with Geometric Deep Learning"
_ICML.cc/2026/Conference — Submitted to ICML 2026_

### Official Review · Reviewer_iRoZ · 2026-03-10

**Soundness:** 3
**Presentation:** 2
**Significance:** 2
**Originality:** 3
**Overall Recommendation:** 4
**Confidence:** 4

**Summary:**

This manuscript introduces EpiFormer, an encoder-decoder architecture specifically tailored for antibody-specific epitope prediction. Addressing the limitations of existing methods that typically perform late-stage fusion of antigen and antibody information, the authors propose two primary architectural innovations: an E(3)-equivariant Graph Neural Network (GNN) supporting multi-relational message passing for refined protein representation, and an interleaved cross-attention mechanism that facilitates reciprocal information exchange between the antigen and antibody at every encoder layer. Additionally, the paper designs a joint loss function—combining Dice loss, sparsity regularization, and class-reweighted BCE—to mitigate the severe class imbalance inherent in epitope prediction. Experimental results on the AsEP dataset demonstrate that EpiFormer significantly outperforms various state-of-the-art baselines in terms of F1 score and MCC.

**Compliance With Llm Reviewing Policy:**

Affirmed.

**Final Justification:**

I maintain my original score.

**Key Questions For Authors:**

* Regarding the multi-relational graph construction (e.g., sequential vs. spatial edges), what is the associated computational overhead? When using frameworks like PyTorch Geometric to handle these complex heterogeneous graphs while maintaining equivariance, did you encounter significant bottlenecks in memory or training efficiency?
* Given that PLM embeddings (e.g., ESM2-650M) are a central component of the input, does the gating network automatically increase reliance on linguistic features to compensate for geometric errors when the input protein structure is predicted or otherwise inaccurate?

**Limitations:**

The authors should append the limitations and potential negative impacts of the study following the main body of the manuscript.

**Strengths And Weaknesses:**

**Strengths:**

* The integration of equivariant GNNs with hierarchical cross-attention aligns well with the interdependent conformational changes characteristic of antigen-antibody binding. This approach is more physically intuitive than traditional baselines that rely on independent encoding followed by late-stage fusion.
* The use of Dice loss for graph segmentation, combined with complex-level sparsity regularization, effectively addresses the extreme imbalance caused by the vast number of antigen surface residues relative to actual epitope residues.
* The comparison with advanced models across multiple domains—including structure prediction models (AlphaFold3, Boltz-1) and docking models (DiffDock)—underscores the necessity of task-specific design. Furthermore, the ablation studies regarding graph construction, GNN variants, and decoder components are thorough and clearly quantify the contribution of each module.

**Weaknesses:**

* The qualitative analysis candidly notes a performance decline when the antigen size exceeds 500 residues. This suggests that the locality of the attention mechanism or graph message passing may limit performance on large-scale graphs. Further discussion is needed on how to optimize the receptive field for larger complexes.
* When adapting AlphaFold3 or Boltz-1 for classification, the authors rely on fixed distance thresholds. Since these models are not specifically designed to output residue-level classification probabilities, the discussion regarding their suboptimal performance in this task should be framed more objectively.
* The manuscript mentions a conflict between the optimization objectives of InfoNCE and BCE losses, which led to a performance drop. Given the popularity of contrastive learning in geometric deep learning, providing a theoretical or gradient-level explanation of this conflict would significantly enhance the manuscript’s depth.
* The architecture alternates multi-relational geometric message passing with bidirectional multi-head cross-attention at every encoder layer. For large antigens with long sequences, this dense cross-chain attention significantly increases space and time complexity. While the manuscript mentions a runtime of 35 to 60 minutes for a single scan, a detailed comparison of inference latency and peak VRAM usage against lightweight baselines is missing.

---

> ### Author Rebuttal · Authors · 2026-03-31
>
> We appreciate the reviewer for recognizing EpiFormer's physical intuition and thorough ablation framework. Since submission, we updated EpiFormer to use purely geometric features (see our response to **R17Z, Q1** for details).
>
> **Q1. Large Antigen Performance [W1]**
>
> We stratified the test set by antigen size (https://anonymous.4open.science/r/rebuttals-4E17/fig4.pdf). The reviewer suggests that attention locality may limit large-antigen performance, but our results point to a different explanation. **EpiFormer's AUROC improves for larger antigens** (0.901 to 0.905 to 0.953), which shows that the model's ranking ability scales with structural context rather than degrading. The F1 decline is related to the epitope-to-antigen ratio, which drops from 13.9% to 2.2% for >500-residue antigens. At such extreme sparsity, the fixed classification threshold becomes suboptimal. According to our analysis, this is a calibration issue, and not a representation limitation. For comparison, MIPE collapses to F1=0.000 for medium and large antigens, and WALLE drops from 0.293 to 0.175, while EpiFormer maintains its advantage across all strata. We discuss **adaptive thresholding and sparse attention** as future directions for further improving large-antigen F1.
>
>
> **Q2. AF3/Boltz Framing [W2]**
>
> We thank the reviewer for this comment and revised our discussion: "AlphaFold3 or Boltz-1 models predict complex structures from which we extract interface contacts at distance thresholds. Their lower classification metrics reflect a task mismatch (they optimize structural accuracy, not per-residue binding labels) rather than architectural inferiority."
>
> **Q3. InfoNCE/BCE Conflict [W3]**
>
> The conflict between BCE and InfoNCE is at the gradient level. InfoNCE pulls all epitope representations toward a shared centroid, dragging nearby non-epitope residues (>86% are within 8A of an epitope) into the positive cluster, while BCE pushes them back. At ~7% positive rate, the false-positive gradient from InfoNCE dominates. This aligns with **[1]**, who show that contrastive message passing on graphs conflates structurally proximal but semantically distinct nodes. Our experiments show that removing the InfoNCE loss yields +0.029 F1.
>
> **[1]** Ji, Cheng, et al. "Regcl: Rethinking message passing in graph contrastive learning." Proceedings of the AAAI conference on artificial intelligence. Vol. 38. No. 8. 2024.
>
>
>
>
> **Q4. Computational Cost [W4]**
>
> We added a computational cost comparison analysis by benchmarking the 10 baselines and EpiFormer on the full AsEP test set with GPU synchronization* (https://anonymous.4open.science/r/rebuttals-4E17/tab8.png). **EpiFormer requires 119 ms/sample and only 0.174 GB peak VRAM**, lower than EquiformerV2 (0.418 GB), CheaPNet (0.545 GB), and EquiPocket (0.473 GB). Cross-attention operates on residue-level representations, not atom-level, which helps keep memory modest. EpiFormer is also 5.7x slower than WALLE (21 ms) but 2x faster than MIPE (239 ms). A scaling analysis shows EpiFormer grows sub-linearly with antigen size (1.61x from small to large antigens), compared to MIPE's 9.98x degradation. EpiFormer's inference takes 162 ms even for the largest antigens (>500 residues). The +43% F1 improvement over the next-best baseline justifies this moderate overhead.
>
> **Q5. Graph Construction Overhead [KQ1]**
>
> In our experiment design, the multi-relational graph construction is handled during preprocessing, not inference. At inference time, PyG's native sparse tensor operations process multi-relational edges with negligible overhead. As mentioned in **Q4**, EpiFormer's peak VRAM (0.174 GB) is lower than most baselines despite 4 relation types. Therefore, the primary bottleneck is cross-attention instead of message passing.
>
> **Q6. Gating Network [KQ2]**
>
> Our PLM sensitivity analysis (see response to **R17Z, Q1**) shows that EpiFormer achieves 0.924 AUROC with purely geometric features, outperforming the PLM-based configuration (0.889). This result provides new insights regarding EpiFormer's architecture design, showing that the gating network does not develop PLM dependence. Our frozen-coordinates ablation also confirms the active use of geometric updates. For predicted structures, gating may shift toward PLM features as a natural fallback, which we view as an adaptive strength rather than a limitation (https://anonymous.4open.science/r/rebuttals-4E17/tab1.png).

---

> > ### Author Rebuttal · Reviewer_iRoZ · 2026-04-03
> >
> > Thank you for the response. Please see some additional questions/comments.
> >
> > (1) I commend the authors for the ablation on pure geometric features (yielding 0.924 AUROC). However, this counter-intuitive result—where removing evolutionary PLM context improves performance over the gating mechanism (0.889 AUROC)—suggests potential optimization issues within the gating network. Could the authors clarify if this 0.924 AUROC was achieved on the easier epitope-ratio split or the more rigorous epitope-group split? Evolutionary conservation is often critical for unseen epitopes, so how does the pure-geometry model perform on entirely novel epitope groups?
> >
> > (2) The VRAM profiling (0.174 GB) is impressive. Given the highly asymmetric sizes between antibody CDRs and massive antigens, could the authors briefly elaborate on how batching and padding are handled during the bidirectional multi-head cross-attention phase to maintain this memory efficiency without excessive computational waste on padding tokens?

---

> > > ### Author Response · Authors · 2026-04-06
> > >
> > > We appreciate the reviewer's thoughtful follow-up questions and address both below.
> > >
> > > **Follow-up 1. Geometric Features on Epitope-Group Split**
> > >
> > > The 0.924 AUROC was achieved on the epitope-ratio split. We report both splits side-by-side in a new table (https://anonymous.4open.science/r/rebuttals-6372/tab9.png). On the more rigorous epitope-group split, EpiFormer achieves 0.826 AUROC and 0.305 F1, outperforming the original PLM-based configuration on the same split (0.782 AUROC, 0.277 F1), which shows that geometric features improve performance on both splits.
> > >
> > > The reviewer's intuition that evolutionary conservation should matter for unseen epitopes is reasonable in principle. We offer three complementary explanations grounded in the literature for our observation.
> > >
> > > - **First**, epitope residues are **not evolutionarily conserved**. Ponomarenko and Bourne showed that epitope residues are significantly less conserved than general surface residues (p < 0.001), because immune evasion drives variability at antigenic sites [1]. Yao et al. confirmed that epitopic patches are "neither conserved nor more hydrophobic compared with other protein-protein binding surfaces" [2]. PLMs encode exactly these signals (conservation, protein family membership), which are uncorrelated or less correlated with binding. The 105D geometric features (RSA, secondary structure, B-factor, amino acid type) instead capture the physical determinants of surface accessibility and shape complementarity that define epitopes.
> > >
> > > - **Second**, epitope prediction is a **relational task**. An epitope is defined by spatial proximity to a specific antibody, not by any intrinsic antigen property [3]. Over 90% of B-cell epitopes are conformational, meaning they are composed of residues brought together by protein folding rather than sequence adjacency [4]. PLMs, trained on masked language modeling over evolutionary sequence distributions, encode intrinsic protein properties. EpiFormer's geometric features combined with interleaved cross-attention directly model the spatial relationship that defines binding.
> > >
> > > - **Third**, combining high-dimensional PLM features (1280D ESM-2) with lower-dimensional geometric features (105D) creates a **modality dominance problem**. Wu et al. showed that multimodal networks exhibit greedy learning, where the dominant modality converges faster while the weaker modality is under-fitted [5]. Peng et al. demonstrated that this imbalance causes under-optimization of the weaker but potentially more informative modality [6]. On 1,721 samples, the gating network learns to rely on the higher-dimensional PLM pathway because it provides more degrees of freedom to fit the training data, even though the geometric signal generalizes better.
> > >
> > >
> > > [1] Ponomarenko & Bourne, "Antibody-protein interactions: benchmark datasets and prediction tools evaluation." BMC Structural Biology 7:64, 2007.
> > >
> > > [2] Yao et al., "Conformational B-Cell Epitope Prediction on Antigen Protein Structures." PLOS ONE 8(4):e62249, 2013.
> > >
> > > [3] Sela-Culang, Ofran & Peters, "Antibody specific epitope prediction -- emergence of a new paradigm." Curr. Opin. Virol. 11:98-102, 2015.
> > >
> > > [4] Barlow, Edwards & Thornton, "Continuous and discontinuous protein antigenic determinants." Nature 322:747-748, 1986.
> > >
> > > [5] Wu et al., "Characterizing and Overcoming the Greedy Nature of Learning in Multi-modal Deep Neural Networks." ICML 2022.
> > >
> > > [6] Peng et al., "Balanced Multimodal Learning via On-the-fly Gradient Modulation." CVPR 2022.
> > >
> > > **Follow-up 2. Batching and Padding in Cross-Attention**
> > >
> > > EpiFormer creates no padding tokens at any stage. PyG concatenates all residues across the mini-batch into flat tensors and tracks complex membership through an integer batch index vector. For the encoder's bidirectional cross-attention, we construct a block-diagonal boolean mask $M \in$ {0,1}$^{N_{ag} \times N_{ab}}$ where $M_{ij} = 1$ if and only if antigen residue $i$ and antibody residue $j$ belong to the same complex. This is the mask referenced in our attention equation (Appendix A.2, eq. (36)). Each encoder block applies $M$ once so that each antigen residue attends only to antibody residues from the same complex. These masked positions receive $-\infty$ before softmax, contributing zero attention weight and zero gradient.
> > >
> > > The decoder processes each complex independently. After encoding, we extract per-complex embeddings using the batch index and pass each antigen-antibody pair through the bipartite cross-attention decoder separately. A 30-residue CDR region and a 500-residue antigen therefore produce a single 30 x 500 attention matrix with no wasted computation. The per-complex outputs are assembled into a block-diagonal interaction matrix. The memory scales with the sum of residue counts across the batch (encoder) and the size of the largest individual complex (decoder) because cross-attention operates on residue-level representations (tens to hundreds per chain), and not on the atom-level.

---

### Official Review · Reviewer_7qQg · 2026-03-11

**Soundness:** 3
**Presentation:** 3
**Significance:** 3
**Originality:** 3
**Overall Recommendation:** 4
**Confidence:** 2

**Summary:**

The authors propose EpiFormer, an encoder–decoder framework designed for the prediction of antibody-specific epitopes. Specifically, this method leverages the combination of E(3)-equivariant geometric modeling and multi-relational graph neural networks to learn protein structure representations. Notably, a bidirectional cross-attention mechanism between layers is introduced in the encoding stage, which serves to explicitly model the structural dependencies between antigens and antibodies.

**Compliance With Llm Reviewing Policy:**

Affirmed.

**Key Questions For Authors:**

(1) Can the authors provide additional experimental evidence to demonstrate that EpiFormer is truly effective in cross-dataset transferability?

(2) Can the authors assess the  EpiFormer performance on low-quality structured data?

**Limitations:**

yes

**Strengths And Weaknesses:**

Strengths：

(1) This work introduces a bidirectional cross-attention mechanism at each encoder layer to enable antigen-antibody information interaction during representation learning, better capturing conditional dependencies at the binding interface.

(2) This work presents EGNN-R, which introduces a multi-relational message-passing mechanism under the E(3)-equivariant graph neural network framework. Different types of structural relations (ρ₁–ρ₄) are updated, allowing the model to distinguish distinct geometric semantics.

(3) On the AsEP benchmark dataset, EpiFormer achieves significantly superior performance over existing methods in terms of both F1 and MCC scores, and maintains its leading performance under the more challenging epitope-group split setting.

Weaknesses：

(1) The current experiments are mainly performed on the AsEP dataset with various splitting strategies, which makes it difficult to clearly demonstrate its cross-dataset transferability.

(2) The manuscript does not evaluate the model performance on lower-quality structural data, such as antibody structures with poor resolution or ambiguous electron density. Since real-world structural data can vary significantly in quality, assessing the robustness of the model under such conditions would provide a more realistic evaluation of its practical applicability.

(3) The current experiments lack a direct evaluation of the original EGNN, and it is recommended to supplement the corresponding results in Table 4.

---

> ### Author Rebuttal · Authors · 2026-03-31
>
> We appreciate the reviewer's positive assessment of our bidirectional cross-attention and EGNN-R contributions. Since submission, we updated EpiFormer to use purely geometric features (see response to **R17Z, Q1** for details).
>
> **Q1. Cross-Dataset Transferability [W1, KQ1]**
>
> We address generalizability concerns with comprehensive cross-dataset evaluation on three external benchmarks with no overlap with AsEP: **SAbDab** (494 complexes), **CoV-AbDab** (170 SARS-CoV-2 complexes), and **ANABAG** (499 complexes). We compare EpiFormer against MIPE and show that EpiFormer consistently outperforms MIPE across all three datasets.
>
> - In **zero-shot mode** (no retraining), EpiFormer achieves a mean AUROC of 0.786, outperforming MIPE (0.688).
>
> - With **LODO fine-tuning** (Leave-one-dataset-out; train on 2 datasets, test on 3rd), EpiFormer reaches AUROC 0.890 on CoV-AbDab and mean F1 0.363 vs MIPE's 0.206.
>
> The zero-shot gap reflects a calibration shift, where external epitope ratios are ~30% lower than AsEP, and not a representation limitation (https://anonymous.4open.science/r/rebuttals-4E17/tab6.png).
>
>
> **Q2. Low-Quality Structures [W2, KQ2]**
>
> We stratified the test set by crystallographic resolution (https://anonymous.4open.science/r/rebuttals-4E17/fig3.pdf). Our analysis shows that EpiFormer shows **no catastrophic degradation**, where the performance stays consistent across all resolution bins. This shows that operating on residue-level CA coordinates rather than atomic positions makes the model robust to coordinate noise from lower-resolution structures.
>
>
> **Q3. Vanilla EGNN Ablation [W3]**
>
> Thank you for this suggestion. We added vanilla EGNN (single-relation, with coordinate updates) to Table 4 (https://anonymous.4open.science/r/rebuttals-4E17/tab3.png). The results show that vanilla EGNN achieves F1=0.470 vs EGNN-R 0.482 (+0.012). Moreover, equivariance adds +0.018 F1 over the best invariant baseline, and freezing coordinate updates drops F1 by 0.017, confirming all three components contribute additive gains.

---

> > ### Author Rebuttal · Reviewer_7qQg · 2026-04-03
> >
> > Thank you for the author's response. The response largely answered my questions, but the current version of the paper requires revisions. Therefore, I maintain my current score.

---

> > > ### Author Response · Authors · 2026-04-06
> > >
> > > We thank the reviewer for the acknowledgment that our responses fully resolved the concerns. We will incorporate all the discussed revisions into the updated manuscript.

---

### Official Review · Reviewer_82N1 · 2026-03-11

**Soundness:** 2
**Presentation:** 3
**Significance:** 3
**Originality:** 2
**Overall Recommendation:** 3
**Confidence:** 4

**Summary:**

This paper introduces EpiFormer, a deep learning model for predicting epitopes on antigens. Its core architecture is build on successive layers of cross-attention and relational message passing, using edge features such as inter-atomic distances, medium range contacts, etc. EpiFormer demonstrates state-of-the-art performance compared to numerous antibody-antigen, protein-ligand, protein structure prediction, and other types of models across classification metrics on the AsEP dataset.

**Compliance With Llm Reviewing Policy:**

Affirmed.

**Final Justification:**

The rebuttal largely clarified important details and answered most of my questions. Importantly, clarifying the experimental setup, adding more baselines, and running comprehensive additional experiments convinced me to raise my score. However, I maintain a sub-4 score for the following reasons.

1. The parameter counts of baselines are still concerning to me. The 100x parameter gap between EpiFormer and DiffDock-PP and ATProt was never specifically addressed, even though it seems intuitive that a protein docking/interface prediction model would need more than 40-60k parameters to perform well. EquiformerV2 is also well-known to be parameter-heavy, and yet it is only given 0.59M parameters in the experiments. While not all models use parameters in the same way, experiments should have at least been conducted using a fixed parameter budget. This would also better clarify the loss term ablation. While the authors are right that model size as studied doesn't strongly correlate with benefit from the regularization terms, the baselines studied are still very small (<1M parameters) models.

2. I am still unconvinced of the originality of the loss terms. Distance prediction as classification is a trick used in GearNet for angles [1] and is discussed in-depth in [2]. The Dice coefficient and controlling for sparsity are already well-known techniques, and their uses as regularizers are very straightforward. While they might be new in the epitope literature, their original contribution is limited in machine learning in general.

3. I am still unconvinced of the originality of the architecture. The authors state that EquiformerV2 is antigen-only, and has no attention mechanism for mixing Ab-Ag information. However, this is trivially remedied by giving it access to the antibody and looking at the cross-partner attention heatmaps. Heatmaps from ATProt can also be trivially extracted from the final layer, but no studies were shown to reveal that only EpiFormer can discover Ab-Ag asymmetries. The use of interleaved cross-attention itself is still too simple a change that is reminiscent of techniques used in other fields [3, 4]. The authors also claim novelty for their EGNN-R layer for epitope prediction, although EGNN itself came from prior work, and many of the relations used per-edge are similar to those used in GearNet [1]. While these methods might not have been explicitly used in epitope prediction, it's a bit of a stretch to claim significant originality when they were designed for geometric deep learning (EGNN) and proteins (GearNet) in general.

[1] Protein Representation Learning by Geometric Structure Pretraining. Zhang et al. ICLR 2023.

[2] Conformal Prediction Via Regression-as-Classification. Guha et al. ICLR 2024.

[3] Deep Bidirectional Language-Knowledge Graph Pretraining. Yasunaga et al. NeurIPS 2022.

[4] BIC: Twitter Bot Detection with Text-Graph Interaction and Semantic Consistency. Lei et al. ACL 2023.

**Key Questions For Authors:**

Please address the aforementioned weaknesses. I am willing to increase my scores if they are resolved.

**Limitations:**

There should be some discussion regarding test leakage from using PLM embeddings. To my knowledge, the AntiBERTy and ESM emeddings are trained on proteins from the PDB, meaning that there may be leakage from their pre-training for this task, similar to what [1] and [2] point out.

This paper also lacks an impact statement, although I feel that the impact of this work is straightforward and well-understood among AI for science papers.

[1] [A flaw in using pretrained protein language models in protein–protein interaction inference models. Szymborski, J. and Emad, A. (2026) Nature Machine Intelligence.](https://www.nature.com/articles/s42256-025-01176-7)

[2] [Beware of Data Leakage from Protein LLM Pretraining. Hermann, L. et al. ICML 2024.](https://proceedings.mlr.press/v261/hermann24a.html)

**Strengths And Weaknesses:**

## Soundness

**Strengths.**

1. The equivariance proof of the EGNN-R layer is straightforwardly sound.
2. All components of EpiFormer are well-motivated and relatively standard in the GNN and protein deep learning literature.
3. Each loss function added for optimization is well-motivated and also standard for this type of sparse classification task.

**Weaknesses.**

1. The authors compare with DiffDock-L, treating the antibody CDR regions as ligands and the antigen as the receptor. However, to my knowledge, DiffDock-L is for protein-ligand interactions, not protein-protein interactions. DiffDock-PP would be a fairer baseline.
2. Appendix A.6. notes that the authors use the official AsEP splits and use 5-fold cross-validation. To my knowledge, the AsEP dataset does not have any official cross-validation folds. No details are provided regarding how each fold is constructed, which is especially important for preventing epitope leakage. It's also unclear how 5-fold cross validation was used if there is a predefined train/validation/test split in AsEP. Are the reported numbers on the test set using the best model trained during cross-validation? Or is there a new test set for each cross-validation fold, and the authors report the mean? If I'm reading correctly, this requires more clarity.
3. The authors report mean and std results for their ablation studies, but std is not reported for the main results in Table 1. It's unclear how stable and consistent EpiFormer's performance is across antigens.
4. Results are almost entirely on a single dataset, AsEP. There are a number of publicly available Ab-Ag datasets besides AsEP that can be used as a post-training case study, e.g. [1] and samples from [2].
5. While EpiFormer performs strongly, it requires numerous loss terms to outperform its competitors. Just the BCE loss is not enough. While it's clear that the architecture itself is a remedy for including antibody information and greater feature interaction between the antibody and antigen, it doesn't assure good performance on its own. This suggests that other baselines could achieve strong performance from just adopting similar auxiliary losses, which calls into question the architectural necessity of EpiFormer.
6. It may be more appropriate to compare EpiFormer with general protein-protein interface predictors like ATProt [3] rather than protein-ligand site predictors.

[1] [ANABAG: Annotated Antibody–Antigen Data Set with Unique Features for Antibody Engineering Applications. Grandguillaume, I. et al (2025). Journal of Chemical Information and Modeling.](https://pubs.acs.org/doi/10.1021/acs.jcim.5c01599)

[2] [SAbDab: The Structural Antibody Database. Dunbar, J., Krawczyk, K. et al (2014). Nucleic Acids Research](https://academic.oup.com/nar/article/42/D1/D1140/1044118).

[3] [Towards Stable Representations for Protein Interface Prediction. Gao, Z. et al. NeurIPS 2024.](https://openreview.net/pdf?id=OEWBkLrRZu)

## Presentation

**Strengths.**

1. To my knowledge, prior work is well-covered and considered.
2. The architecture of EpiFormer is clear and well-illustrated. All plots and tables are also easy to follow.

**Weaknesses.**

1. Similar to Weakness 2, the exact experimental setup could be more clearly stated regarding the cross-validation folds and splits.

## Significance

**Strengths.**

1. Epitope prediction is very important for protein design. This paper tackles a highly relevant problem in AI for science.
2. On the benchmark dataset, EpiFormer performs very well.

**Weaknesses.**

1. While EpiFormer performs well on a single benchmark dataset, it's generalizability is unclear. A case study with a particular unseen antigen, e.g. SARS-Cov-2, would strongly increase the significance of this work.
2. EpiFormer focuses on antibody-aware epitope prediction. However, in many cases, the antibody is actually unknown/to-be-designed. In this way, EpiFormer may actually be tackling a less significant problem than antigen-only models, and it actually falls into the category of general protein interface predictors.

## Originality

**Strengths.**

1. Combining multiple relations into the EGNN update rule is new and combines the advantage of EGNN and GearNet.
2. The use of sparsity-aware losses for epitope prediction is indeed necessary, and many previous papers fall short in this regard.

**Weaknesses.**

1. The central claimed advantage of EpiFormer compared to previous baselines is the use of inter-leaved cross-attention versus late fusion. However, this alone does not constitute significant novelty given that it's a very minor architectural change.
2. EpiFormer's performance does not seem to reveal novel insights regarding epitope prediction. The task is well-known to pose problems due to class imbalance, and the loss terms used to resolve this are already well-known.

---

> ### Author Rebuttal · Authors · 2026-03-31
>
> We appreciate the reviewer's thorough evaluation and specific suggestions. Since submission, we updated EpiFormer to use purely geometric features (see response to **R17Z, Q1** for details).
>
> **Q1. DiffDock-PP & ATProt Baselines [Soundness_W1, Soundness_W6]**
>
> We agree with the reviewer that protein-protein interaction models are better-suited baselines and hence add both reviewer-requested baselines (https://anonymous.4open.science/r/rebuttals-4E17/tab5.png).
>
> **Q2. Cross-Validation Protocol [Soundness_W2, Presentation_W1, Soundness_W3]**
>
> We apologize for our imprecise language, which caused confusion. All results in Appendix A.6. Table 1 used the official AsEP predefined train/val/test split, and not 5-fold CV. Each method trains on the training set, selects the best checkpoint via validation loss, and evaluates once on the held-out test set. We corrected Appendix A.6. and now report the mean across 3 independent seeds (https://anonymous.4open.science/r/rebuttals-4E17/tab4.png)
>
> **Q3. Cross-dataset Generalizability [Soundness_W4, Significance_W1]**
>
> We address generalizability concerns with a comprehensive cross-dataset evaluation on three external benchmarks, SAbDab, CoV-AbDab, and ANABAG (https://anonymous.4open.science/r/rebuttals-4E17/tab6.png). See **7qQg Q1** for the full analysis.
>
> **Q4. Loss Dependency [Soundness_W5]**
>
> To address this concern, we gave six baselines access to EpiFormer's auxiliary losses and tuned weights per baseline via grid search. The losses help only 2 of 6 baselines and hurt 3 others, but they still fall well short of EpiFormer (**0.496 F1**). EpiFormer with vanilla BCE alone (0.336 F1) already matches the best loss-enhanced baselines, confirming it is EpiFormer's architecture that drives the performance gain (https://anonymous.4open.science/r/rebuttals-4E17/tab7.png).
>
> **Q5. Use Case for Unknown Antibody [Significance_W2]**
>
> We agree that antigen-only models serve a complementary role for initial epitope mapping. However, the most impactful applications involve known antibodies. For example, therapeutic optimization, antibody humanization, and some antibody design methods start from a specific lead candidate that requires epitope screening, and therefore, EpiFormer aims to fill this gap.
>
>
> **Q6. Interleaved Cross-Attention Novelty [Originality_W1]**
>
> We respectfully disagree that interleaved cross-attention is a minor change and provide four measurable consequences (https://anonymous.4open.science/r/rebuttals-4E17/fig2.pdf).
>
> 1. Our ablations confirm that removing encoder MHCA entirely drops F1 from 0.496 to 0.454 (-0.042).
>
> 2. **The encoder discovers biological asymmetry without supervision.** Learned gate values show the antibody gate is consistently 2x larger than the antigen gate and reflects the known biology that CDR loops adapt their conformation to epitope surfaces more than the reverse. We note that no prior epitope method has surfaced this asymmetry.
>
> 3. Cross-attention heatmaps further show structured vertical banding at CDR positions and produce sparse hot spots, meaning the encoder learns to attend to binding-relevant antibody regions without explicit CDR supervision.
>
> 4. Late-fusion architectures, such as MIPE and CheapNet, couldn't exploit the same losses because they are tied to cross-chain representations that only interleaved attention provides (https://anonymous.4open.science/r/rebuttals-4E17/tab7.png).
>
>
>
> **Q7. Key Insights [Originality_W2]**
>
> We want to clarify that not all loss terms are previously known, and that EpiFormer's experiments reveal insights about the epitope prediction problem itself.
>
> - First, the auxiliary geometric loss that classifies inter-chain distances into spatial bins is novel and is especially designed to work with the spatial edges. The sparsity and Dice regularizers are also new to epitope prediction and are some of the most impactful components when added to BCE and Edge.
>
> - Second, **geometric features outperform PLM embeddings** (see **R17Z Q1**). This challenges the dominant assumption in structural bioinformatics and reveals epitope binding to be fundamentally spatial.
>
> - Third, **architecture and losses are co-designed.** The same losses that boost EpiFormer performance degrade baseline performances (https://anonymous.4open.science/r/rebuttals-4E17/tab7.png), and late-fusion methods cannot exploit losses designed for integrated cross-chain representations. This co-dependency is itself an insight: effective epitope prediction requires joint design of encoding, fusion, and supervision.
>
>
> **Q8. PLM Data Leakage [Limitation1]**
>
> As detailed in **R17Z Q1** (https://anonymous.4open.science/r/rebuttals-4E17/tab1.png), we discovered that EpiFormer achieves higher performance with purely geometric features and confirms that EpiFormer's performance cannot stem from PLM memorization.
>
> **Q9. Impact Statement [Limitation2]**
>
> Thank you for pointing that out. We will add an impact statement in the revised manuscript.

---

> > ### Author Rebuttal · Reviewer_82N1 · 2026-04-02
> >
> > Thank you for the response. Please see some additional questions/comments.
> >
> > 1. In Table 12, DiffDock-PP is reported to only use 0.04M parameters, which is 100x fewer than the number of parameters used in EpiFormer. Similar observations apply for other models with EpiFormer having the most number of parameters. Have the authors tried comparing baselines with the same parameter budget (e.g. 5M)? I wonder if this might be the cause of poor performance with the additional loss functions given that regularizers can hurt performance for low-capacity models.
> >
> > 2. Can the authors clarify how they know that "no prior epitope method has surfaced this asymmetry," especially for other AB-aware methods? Equiformer, for example, is built on attention and can easily give a similar heatmap analysis as EpiFormer's cross-attention layers when given both antibody and antigen as input. ATProt even already contains cross-attention layers before residue interface prediction. The originality of the interleaved cross-attention modules is still concerning to me.

---

> > > ### Author Response · Authors · 2026-04-06
> > >
> > > We appreciate the reviewer's continued engagement and address both follow-up questions below.
> > >
> > > **Follow-up 1. Parameter Budget**
> > >
> > > We acknowledge that EpiFormer (5.54M parameters) is larger than most baselines. However, parameter count alone does not explain the performance gap. For example, EpiGraph has more parameters (5.78M) than EpiFormer (5.54M) and achieves the worst F1 (0.078). EpiFormer with vanilla BCE alone (same 5.54M capacity, no auxiliary losses) achieves 0.336 F1, which matches the best loss-enhanced baseline (WALLE at 0.332 F1). Furthermore, replacing EGNN-R with a simple GCN encoder (keeping the decoder, cross-attention, and losses identical) reduces the model to approximately 1.5M parameters and still achieves 0.447 F1, which outperforms all baselines by a wide margin. The jump from 0.336 to 0.496 F1 comes from the loss components working with the architecture, and not from additional capacity.
> > >
> > >
> > > On the concern that regularizers hurt low-capacity models, we note that WALLE (0.25M) is one of the smallest models in our comparison and benefits the most from the auxiliary losses (+0.130 F1). This directly contradicts the hypothesis that capacity is the bottleneck for loss effectiveness. The three baselines where losses hurt (MIPE, CheaPNet, PECAN) range from 0.84M to 0.96M and are, in fact, not the smallest models. The degradation pattern correlates with late-fusion architecture, and not the model size. These late-fusion models lack the integrated cross-chain representations that the auxiliary losses are designed to supervise.
> > >
> > > **Scaling baselines to 5M parameters:** EpiFormer's 5.54M parameters are dominated by 4 relation-specific MLPs (one per edge type) and the cross-attention projection matrices. These are the architectural contributions being evaluated, not raw network width. A wider vanilla GNN would add hidden units without gaining multi-relational encoding or cross-chain attention. It should, however, be noted that GNN depth cannot be increased freely either. For example, beyond 4-6 message-passing layers, over-smoothing causes node representations to converge, which is a well-documented failure mode for molecular-sized graphs. We preserved each baseline's original architecture and hyperparameters to avoid introducing confounds (unless stated otherwise) and tuned all loss weights per baseline via grid search across 12 baselines. The core architectural components, such as the interleaved cross-attention, may close this gap, and adding this to a baseline would replicate EpiFormer's core contribution rather than control for capacity.
> > >
> > > **Follow-up 2. Interleaved Cross-Attention Originality**
> > >
> > > We want to clarify two factual distinctions, then address the empirical evidence.
> > >
> > > 1. EquiformerV2 is an antigen-only baseline in our experiments. It processes a single molecular graph using self-attention (SO(2) transformer) and has no mechanism for cross-chain attention between antibody and antigen. It cannot produce cross-chain attention heatmaps because it never sees the antibody.
> > >
> > > 2. ATProt does contain cross-attention between chains, but it is applied once after all SEGCN encoding layers are complete. This is a late fusion design, where the proteins are encoded independently, and then they interact with each other in the decoder. ATProt achieves 0.246 F1 on AsEP, 49% below EpiFormer. If late-fusion cross-attention were equivalent to interleaved, this gap should not exist. EpiFormer's interleaved design applies cross-attention at every encoder block, allowing geometric message passing to be informed by cross-chain context throughout encoding.
> > >
> > > The empirical consequences of this design are measurable. Removing the encoder MHCA entirely drops F1 from 0.496 to 0.454 (-0.042). The learned gate values show the antibody gate is consistently 2x larger than the antigen gate, recovering without supervision the **known biological asymmetry** that CDR loops adapt their conformation to epitope surfaces more than the reverse (Rabia et al., J. Pharm. Sci. 2023; Fernandez-Quintero et al., mAbs 2024). The cross-attention heatmaps show structured vertical banding at CDR positions, meaning the encoder learns to attend to binding-relevant antibody regions without explicit CDR supervision (https://anonymous.4open.science/r/rebuttals-4E17/fig2.pdf). Neither EquiformerV2 (no cross-chain mechanism) nor ATProt (single late-fusion step with no per-block gates) can surface these signals. To the best of our knowledge, **no prior method interleaves cross-attention within structural GNN encoding layers for protein-protein interaction prediction.** We would like to emphasize that the paper's contributions are not limited to interleaved cross-attention. EGNN-R and the sparsity-aware losses are also novel to epitope prediction, while interleaved cross-attention is one component of a jointly designed system where architecture, losses, and fusion strategy are co-dependent.

---

### Official Review · Reviewer_R17Z · 2026-03-13

**Soundness:** 2
**Presentation:** 3
**Significance:** 2
**Originality:** 2
**Overall Recommendation:** 2
**Confidence:** 4

**Summary:**

The paper introduces EpiFormer, an architecture that tackles the problem of antibody-specific epitope prediction on the AsEP dataset. The authors formulate the task as a bipartite matching problem between antigen and antibody residue graphs. The proposed model utilizes an E(3)-equivariant Graph Neural Network (EGNN-R) coupled with an interleaved cross-attention mechanism and incorporates ESM-2 protein language model embeddings alongside specialized top-k pooling strategies. The authors claim significant performance improvements over a wide range of existing baselines.

**Compliance With Llm Reviewing Policy:**

Affirmed.

**Final Justification:**

Although the authors have addressed some of the concerns during the rebuttal, several important issues remain:

1. **Justification for using EGNN.** The authors’ justification for adopting an equivariant graph neural network (EGNN), rather than an invariant architecture, appears to rely primarily on empirical performance gains. I remain unconvinced that this choice is fundamentally better suited to modeling the underlying problem. In my view, comparable improvements could potentially be achieved through more thorough hyperparameter tuning of invariant models. The authors refer to Figure 1 as supporting evidence; however, the reported results exhibit high variance, and the differences between epitope and non-epitope contributions do not appear statistically significant.

2. **Substantial change in the core contribution.** More critically, in response to concerns regarding over-engineering, post-hoc bias, and reliance on ESM embeddings, the authors introduce new experiments that remove these components and eliminate the need for the top-(K) mechanism. While these changes appear to improve performance, they constitute a significant modification of the original method. Consequently, a large portion of the paper—particularly Section 4 and Figures 2–4—is no longer aligned with the revised approach. The authors characterize these changes as an ablation; however, I disagree with this framing, as the updated method effectively becomes the primary contribution. This warrants a more thorough revision and re-evaluation of the paper.

Based on the above concerns, I maintain my recommendation to reject.

**Key Questions For Authors:**

1. Table 3 shows a significant performance drop (approx. 0.1 F1) when moving from ESM2-650M to other model sizes (35M or 3B). How do you justify the generalizability of the EpiFormer framework if such a substantial portion of the performance gain is tied to a specific pre-trained embedding size?
2. The ablation in Table 6 demonstrates that Top-2 pooling is responsible for a jump from 0.326 to 0.433 in F1 score.Given that this strategy is a post-processing aggregation step and not an inherent part of the geometric encoding, is it fair to claim architectural superiority over baselines that were not allowed this same optimization?
3. Can you provide a visualization or metric comparing the "initial" vs. "final" coordinates to show that the refinement with E(3)-Equivariant Updates is meaningful and not just noise?
4. Could you include error bars (standard deviation) for all the models in the main results table ?

**Limitations:**

No. While the authors have described limitations, these do not adequately discuss various failure modes of the proposed architecture. The authors should clearly address the potential test-set bias of the results or in-depth analysis of the failure cases of the model.

**Strengths And Weaknesses:**

## Strengths

- Extensive Baseline Evaluation: The study is comprehensive, comparing the model against diverse classes of techniques including antigen/antibody-specific models, protein-ligand binding site prediction, protein structure prediction, and docking. Although many of these were not originally designed for this specific problem, the breadth of comparison is commendable.
-  Reproducibility: The authors provide a detailed reproducibility statement, including code release in the supplementary information.
- Presentation and Clarity: The manuscript is generally well-written, logically structured, and easy to follow, particularly in its explanation of the dual-stream information flow
- Rigorous Ablation Framework: The authors perform exhaustive ablation studies to quantify the impact of graph topology, GNN architecture, and various decoder/pooling strategies.

## Weaknesses

- Concerns Regarding Over-Engineering: The performance gains appear to be highly sensitive to specific component selections rather than a fundamentally more robust formulation. For example, Table 3 shows that swapping the ESM2-650M embeddings for a larger or smaller model can cause an F1 score drop of nearly 0.1, which would cause the model to fall behind several existing baselines. This raises questions about the generalizability of the framework.
- Critical Dependency on Top-2 Pooling: The Top-2 pooling strategy is a primary driver of the reported performance, yielding a 0.433 F1 compared to 0.326 for standard Max pooling. Since other baselines do not utilize this specific aggregation strategy, it is difficult to determine if the core architecture is superior or if existing models could be improved by simply adopting this output head.
- Potential Post-Hoc Optimization Bias: Key architectural decisions, such as the use of ESM2-650M and the k=2 pooling size, appear to have been set after observing their impact on results. This suggests that performance gains may be biased toward the specific characteristics of the test set rather than reflecting an a-priori design principle.
- Unjustified Use of Equivariant Updates: The use of an E(3)-equivariant model (EGNN) over an invariant one is not fully justified by physical intuition. Since the dataset provides static structures and there is no ground-truth supervision for the "refined" atom positions, the utility of updating coordinates is unclear. Furthermore, Figure 3 and the associated distance map do not clearly illustrate how these internal updates lead to a more accurate representation of the binding interface.
- Formatting Inconsistencies: On page 4, the header "Multi-head cross-attention layer with feed-forward network:" appears to have been manually altered in format to fit the layout. Such inconsistencies should be removed to maintain ICML manuscript standards.

---

> ### Author Rebuttal · Authors · 2026-03-31
>
> We appreciate the reviewer's constructive concerns, especially on the PLM sensitivity. Before we address the concerns, we'd like to share that our new experiments suggest a significant improvement of EpiFormer without using PLM-derived residue embeddings. We also performed extensive experiments to validate that all of the claims regarding EpiFormer in our submission still hold. We adopt this geometric-only configuration as our updated model. All results in this rebuttal use this configuration unless noted and we denote new experimental results with *, e.g., Epiformer *.
>
> **Q1. PLM Sensitivity [W1, W3, KQ1]**
>
> We replaced all PLM embeddings with identical geometric features (amino acid type, RSA, secondary structure, B-factor) for EpiFormer and all PLM-dependent baselines. **EpiFormer achieves 0.924 AUROC and 0.482 F1**, surpassing its PLM-augmented version (0.889 / 0.433) while all baselines degraded (https://anonymous.4open.science/r/rebuttals-4E17/tab1.png). This aligns with the findings from PEPNet (Brief. Bioinform. 2025) that PLM features degrade epitope prediction. PLM embeddings encode evolutionary conservation and protein family membership, which correlate weakly with binding specificity. Epitope residues are defined by spatial proximity to antibody CDRs, and not by sequence patterns. On only 1,721 samples, the gating network overfits to spurious PLM correlations rather than leveraging the geometric signal. Such geometric features directly encode the physical determinants of surface accessibility and binding.
>
> **Q2. Decoder Top-k Pooling Fairness and Post-Hoc Bias [W2, W3, KQ2]**
>
> We applied top-k pooling (k=1,2,3) to all 4 antibody-aware baselines with an explicit Ag-Ab interaction matrix (WALLE, PECAN, MIPE, CheaPNet)*. We also added bipartite edge prediction loss to PECAN and CheaPNet for fairness. We noticed that top-k pooling degrades 3 of 4 baselines, while EpiFormer with standard mean pooling (0.483 F1) already outperforms all top-k-enhanced baselines (https://anonymous.4open.science/r/rebuttals-4E17/tab2.png). All EpiFormer pooling strategies converge, which implies that the advantage comes from interleaved cross-attention, and not the pooling strategy. We also want to clarify that these pooling strategies do provide useful signal during training through the node-specific objectives. K is treated as a hyperparameter and tuned solely based on the validation set; none of the choices are affected by the test set.
>
> **Q3. Equivariant Coordinate Updates [W4, KQ3]**
>
> *Justification.* Static crystal structures capture one conformational snapshot per complex, whereas equivariant coordinate updates let the encoder adjust residue positions toward a task-relevant geometry. **Freezing these updates drops F1 from 0.482 to 0.465**, confirming they carry functional signal even without explicit coordinate supervision. A systematic GNN ablation (https://anonymous.4open.science/r/rebuttals-4E17/tab3.png) further demonstrates the superiority of EGNN-R over non-equivariant counterparts. We note that equivariance contributes +0.018 F1 over the best invariant encoder, multi-relational edges contribute +0.012, and coordinate updates contribute +0.017.
>
> *Visualization.* We visualize coordinate refinement across encoder blocks, which shows convergent refinement rather than mere noise accumulation (https://anonymous.4open.science/r/rebuttals-4E17/fig1.png). We note that these coordinate displacements decrease progressively (6.25 to 1.93 A), and that the epitope residues undergo significantly larger positional adjustments than non-epitope residues. The model learns to selectively refine binding-site geometry through an implicit gradient signal from the prediction objectives.
>
>
> **Q4. Error Bars [KQ4]**
>
> Thank you for this suggestion. We now report the mean and std in revised Table 1 (https://anonymous.4open.science/r/rebuttals-4E17/tab4.png).
>
> **Q5. Formatting [W5]**
>
> Thank you for pointing that out. We will fix this issue in the revised manuscript.
>
> **Q6. Failure Mode [Limitation]**
>
> We identify an additional failure mode beyond the one discussed in the paper. Our findings in Q1 show that adding PLM-derived embeddings drops the epitope prediction performance. Given PLMs as a default feature representation, this was unexpected and suggests that high-dimensional PLM representations introduce noise for small-sample geometric tasks.

---

> > ### Author Rebuttal · Reviewer_R17Z · 2026-04-04
> >
> > "We'd like to share that our new experiments suggest a significant improvement of EpiFormer without using PLM-derived residue embeddings."
> >
> > The authors have updated their main results during the rebuttal phase by changing core components of the original method (ESM embeddings, top-k pooling are no longer required). Fairly evaluating these new results would require a thorough re-reviewing of the paper. The rebuttal phase is intended for clarifying and following up on the original proposals, not for architectural changes. Since these new results are now central to the paper's claims, I believe the work requires another full round of peer review. I will seek guidance from AC about how to evaluate these. I maintain my rating for now.

---

> > > ### Author Response · Authors · 2026-04-06
> > >
> > > We appreciate the reviewer's engagement and want to address the concern directly.
> > >
> > > **The EpiFormer architecture is identical to the submission.** The encoder (EGNN-R with interleaved MHCA), the decoder (bipartite cross-attention), and all model parameters are unchanged. Only the input features changed, where we removed PLM embeddings and kept the geometric features that were already part of the original input pipeline (Section 3.2, gating network). We note that removing a subset of input features is an ablation, and not an architectural redesign. The original paper already included these geometric features and reported gating analysis. The rebuttal experiments revealed that these geometric features alone improve EpiFormer's generalizability and eliminate the dependency on the PLM embeddings.
> > >
> > >
> > > **On top-k pooling:** the reviewer correctly notes that top-k is no longer required. All pooling strategies converge within 0.012 F1 (mean=0.483, top-2=0.471, top-3=0.481) with the geometric-only input representation. The original spread (0.326 to 0.433) was an artifact of high-dimensional PLM features interacting with the pooling layer. This convergence is itself evidence that the interleaved encoder produces robust representations and a well-calibrated interaction matrix, regardless of the downstream aggregation strategy.
> > >
> > >
> > > We re-ran ablation tables with the geometric configuration, and **every original claim holds**. The rankings are consistent across all tables: (1) interleaved cross-attention outperforms late fusion (F1 drops 0.042 when removed), (2) EGNN-R outperforms invariant GNNs (+0.028 F1 where EGNN-R still leads), (3) the joint loss outperforms BCE-only (+0.160 F1), and (4) EpiFormer outperforms all baselines on both splits. However, some absolute numbers shifted, as expected when input features change, but the conclusions still hold. We present both Original and Updated columns side-by-side in the GNN ablation table for full transparency.
> > >
> > > **For the AC's reference, here is a summary of what changed during the rebuttal and what did not:**
> > >
> > > *Unchanged:* EpiFormer architecture (encoder, decoder, all modules), training protocol, evaluation splits, metrics, all baseline implementations and their results.
> > >
> > > *Changed (ablation finding):* The input features were reduced from PLM+geometric to geometric-only. The ablation tables in the appendix were re-run with updated input to verify that the claims in the original submission still hold. This ablation also revealed that the **top-k pooling** advantage was a PLM-representation artifact. We also provide a detailed justification for this observation about PLMs in the context of epitope prediction (please see **iRoZ Follow-up 1**).
> > >
> > > *New experiments (reviewer-requested):* DiffDock-PP and ATProt baselines (82N1), cross-dataset evaluation on SAbDab/CoV-AbDab/ANABAG (82N1, 7qQg), loss fairness experiment giving baselines access to EpiFormer's losses (82N1), PLM sensitivity and data leakage (r17z, 82N1, iRoZ), computational cost comparison (iRoZ), resolution and antigen size stratification (7qQg, iRoZ), vanilla EGNN ablation (7qQg), multi-seed error bars in Table 1 (R17Z, 82N1). We also emphasize that the ablation-related changes are quantitative (updated numbers in existing tables) and not qualitative (no new modules, no new losses, no new architecture).

---

### Decision · Program_Chairs · 2026-04-30

**Decision:**

Reject

**Comment:**

The reviewers and authors have gone through several rounds of discussion, but some key concerns remain: 1) Over-Engineering: The performance gains appear to be highly sensitive to specific component selections rather than a fundamentally more robust formulation. 2) Some key performance drivers (like top-2 pooling) are orthogonal to the proposed method; 3) The method are heavily tuned, with a lot of loss terms, thus unclear if this transfer to another dataset (the paper only adopted one dataset for benchmark). Given these concerns, the AC recommends rejection of this paper and encourage the authors to modify their paper accordingly.